# Individual recognition in a jumping spider (*Phidippus regius*)

**Christoph D Dahl[1,2]\*, Yaling Cheng[2]**

[1]Institute of Biology, University of Neuchâtel, Neuchâtel, Switzerland; [2]Graduate Institute of Mind, Brain and Consciousness, Taipei Medical University, Taipei, Taiwan

**\*For correspondence:**
christoph.d.dahl@gmail.com

**Competing interest:** The authors declare that no competing interests exist.

## eLife Assessment

This study provides a **valuable** examination of the social discrimination abilities of a jumping spider, Phippidus regius, based on visual cues. Behavioral essays yielded **solid** evidence that these spiders discriminate between familiar and unfamiliar individuals on the basis of visual cues, however the experimental support for individual recognition and long-term memory is **incomplete**. While the results supply evidence of discrimination, additional experiments would be needed to verify the evidence of individual recognition.

**Abstract** Individual recognition is conceptually complex and computationally intense, leading to the general assumption that this social knowledge is solely present in vertebrates with larger brains, while miniature-brained animals in differentiating societies eschew the evolutionary pressure for individual recognition by evolving computationally less demanding class-level recognition, such as kin, social rank, or mate recognition. Arguably, this social knowledge is restricted to species with a degree of sociality (sensu Wilson, 2000, for a review Gherardi et al., 2012). Here, we show the exception to this rule in a non-social arthropod species, the jumping spider *Phidippus regius*. Using a habituation-dishabituation paradigm, we visually confronted pairs of spatially separated spiders with each other and measured the 'interest' of one spider towards the other. The spiders exhibited high interest upon initial encounter of an individual, reflected in mutual approach behaviour, but adapted towards that individual when it reoccurred in the subsequent trial, indicated by their preference of staying farther apart. In contrast, spiders exhibited a rebound from habituation, reflected in mutual approach behaviour, when a different individual occurred in the subsequent trial, indicating the ability to tell apart spiders' identities. These results suggest that *P. regius* is capable of individual recognition based on long-term memory.

## Introduction

Recognising individuals is a complex cognitive process requiring flexible learning and recognition memory. Arthropod species possessing the social ability of individual recognition would, thus, stand in stark contrast to the commonly accepted notion that animals with smaller brains are cognitively less advanced due to reduced computational power of nervous systems with smaller and fewer neurons (***Niven and Farris, 2012***). And yet, there is evidence for an arthropod species displaying face learning (***Sheehan and Tibbetts, 2011***) and long-term social memory (***Sheehan and Tibbetts, 2008***). That is, a social wasp species (*Polistes fuscatus*) showed mammal-like face learning (***Sheehan and Tibbetts, 2011***; ***Tibbetts, 2002***), arguably providing social benefits by reducing aggression and stabilizing social interactions. As one of the few reported instances of individual recognition in arthropods (see also ***Dreier et al., 2007***), this has contributed to the prevailing view that non-social arthropod species are unlikely to evolve such complex cognitive processes. The underlying reasoning is that individual recognition

**eLife digest** Recognising familiar individuals helps animals navigate their social lives, including finding mates, avoiding rivals, and caring for their young. This skill, called individual recognition, is often linked to social species with large brains, and it supports higher cognitive functions such as empathy, reputation tracking, and attributing mental states.

Recognising an individual requires memory and flexible matching across viewpoints and lighting, and larger brains have more neurons and brain areas that respond strongly to faces. Jumping spiders are tiny hunters with excellent vision, but they are mostly solitary and rarely encounter the same spider twice. Previous studies showed that these animals can distinguish species and sex using colours and patterns. However, it was less known whether a jumping spider can also recognise a specific individual.

To find out more, Dahl and Cheng set up face-to-face meetings between regal jumping spiders (*Phidippus regius*) while keeping them safely separated by transparent panels. Overhead video tracked how close the pair chose to be. Mutual distance was treated as a simple readout of interest, while moving closer suggested stronger attention to the other spider. Keeping distance indicated reduced interest or growing familiarity.

When a spider first saw another individual, it often approached. If the same pair met again minutes later, they tended to keep a bit more distance, consistent with becoming familiar with that specific spider. But when the next encounter featured a different individual, the spiders moved closer, showing increased interest. Across many controlled pairings, this clear behavioural switch between familiar and new repeated reliably.

The effect weakened as the same individuals were shown repeatedly across sessions, even hours later, potentially indicating that the spiders were building up familiarity. Introducing a completely new individual at the end of the experiment – after hours of testing – produced the strongest renewed interest. This late rebound is inconsistent with general fatigue or waning motivation. If spiders were merely tired, an approach would be expected to drop for all stimuli. Instead, this points to sensitivity to individual novelty. In other words, the spiders retain information about who they have seen and retrieve this information after delays – a pattern consistent with encoding and storing individual-specific visual features and comparing new views against that record.

These findings invite a shift in perspective. If a tiny, mostly solitary spider can recognise individuals, how much brain is really needed for flexible social memory? We may underrate animals with compact nervous systems because we treat brain size as a stand-in for cognition. What do such abilities imply for debates about animal consciousness? Behaviour cannot prove subjective experience, but it narrows what kinds of minds are plausible. Future work can map individual recognition across animal groups using the same test. It can then test whether the pattern tracks lifestyle and sensory systems more than brain size, and whether the trait evolved many times independently or from an ancient common origin.

entails high energetic costs, longer processing times, and consequently an increased risk of predation – costs that would not be outweighed by the limited number of social encounters or the marginal survival benefits they provide (*Gherardi et al., 2012*; *Dale et al., 2001*). The general consensus, thus, is that a certain degree of sociality sensu Wilson (*Wilson, 2000*) is required for the emergence of individual recognition (*Dale et al., 2001*). Here, we challenge this consensus by testing for individual recognition in *Phidippus regius*, a non-social, miniature-brained jumping spider. In a controlled experimental procedure, we confronted subjects with live conspecifics, intensifying the social encounter and ensuring ecological relevance in the stimulus presentation. While jumping spiders have been shown to use visual and chemical cues for species, sex, or rival recognition (*Tedore and Johnsen, 2013*; *Cross et al., 2020*), there is no direct evidence for memory-based recognition of individual identity, particularly in taxa such as jumping spiders, where repeated encounters with the same conspecific are infrequent.

## Results

As a first step, we assessed the ability of *P. regius* to individually recognise other members of its species, commonly referred to as individual recognition (*Tibbetts and Dale, 2007*) or individuation of

conspecifics (*Bonatti et al., 2002*). To test this, we employed a habituation-dishabituation paradigm, in which one individual habituates to the extended presence of another individual in its close proximity. Following a brief phase of visual separation, a different individual is introduced. If the focal individual can discriminate the current individual from the former, it is expected to exhibit dishabituation (*Humphrey, 1974*; *Dahl et al., 2007*). In other words, in this habituation-dishabituation paradigm, we expect the rebound in 'interest' to be greater when the identity of the spider changes than when the same individual is presented again. To experimentally control the animal pairs, we placed each individual in a separate container with one transparent side and a transparent top panel. We then pairwise confronted the individuals by placing the containers such that the transparent sides faced each other, allowing the spiders to visually explore and approach one another at close range while preventing any direct physical interaction.

Each trial followed the same procedure: Two spiders, say individuals A and B, were exposed to one another for 7 min, eliciting an initial 'interest' in each other. They were then visually separated for 3 min using an opaque slider. Following this separation, the same pair could be re-exposed to one another (A vs B, *habituation* trial) or either individual could be paired with a new individual (A vs C, or B vs D, *dishabituation* trial), again for 7 min, followed by another 3-min separation period. Relative interest was quantified by approximating spatial distances between the spider pairs in the xy-plane from video recordings captured through the transparent top panel, where high interest is reflected in smaller values (i.e. spiders move closer) and low interest in larger values (spiders maintain distance). Under the assumption that spiders are capable of individuating each other, we predict that in the *habituation* condition, where the same individuals are re-encountered, relative interest decreases, resulting in an increase in distance (*Figure 1a–c*; 'Habituation', hypothetical example 1: spiders seek maximal distance to each other [dashed line]; example 2: spiders seek medium distance to each other [solid line]). Conversely, in the *dishabituation* condition, where a new individual is introduced, relative interest increases and spiders approach each other, leading to a decrease in distance (*Figure 1a–c*; 'Dishabituation'). It is important to clarify the use of the term 'Baseline' as illustrated in *Figure 1a*. The *baseline* trial shown there represents only the first trial in each session, before any habituation or dishabituation has occurred. For all subsequent comparisons, relative changes in proximity were computed by comparing each trial to the immediately preceding one. Specifically, each *dishabituation* trial was analysed relative to the preceding *habituation* trial (*Figure 1c*: 'Dishabituation - Habituation'), while each *habituation* trial (except for the first in each session) was analysed relative to the preceding *dishabituation* trial (*Figure 1c*: 'Habituation - Baseline', equivalent to 'Habituation - Dishabituation'). Thus, habituation and dishabituation are not absolute responses, but are defined by how much the behaviour shifts relative to a reference point. As a result, the difference scores reflect relative changes in the frequency distribution of distances. Some values in the hypothetical distributions (*Figure 1c*) are negative, indicating that the number of times a spider was at a given distance was lower compared to the preceding trial. A decrease in interest can be inferred when this underrepresentation (i.e. negative values) occurs in the more proximal distance bins, accompanied by an overrepresentation (i.e. positive values) in the medium or distal distance bins. Conversely, an increase in interest can be inferred when there is an overrepresentation of occurrences (i.e. positive values) in the proximal distance bins, coupled with an underrepresentation (i.e. negative values) in the medium or distal distance bins. These relative changes in proximity describe the behavioural signature of habituation and dishabituation (*Figure 1c*) and constitute the basis for our statistical analyses.

In the first experiment, we divided a total of 20 individuals into five groups of four individuals each. Each individual of each group was exposed to the three group members in both *habituation* and *dishabituation* trials, resulting in six trials per session, equivalent to one hour of recording. We repeated this procedure twice, resulting in 18 trials across three sessions and an exact repetition of a given trial (and pairing of individuals) in 1 hr intervals (for a detailed description of the procedure see Materials and methods and *Table 1* and *Table 2*).

We found that spiders adjusted their proximity depending on whether they encountered a familiar or a new individual. In *habituation* trials, where the same individual was presented again, spiders tended to maintain greater distances. In contrast, in *dishabituation* trials, where a new individual was introduced, spiders were more likely to stay at close distances. This pattern was statistically robust: *habituation* and *dishabituation* trials (i.e. predictor variable *condition*) differed significantly as a function of inter-individual distances (i.e. predictor variable *distance*), leading to a significant improvement

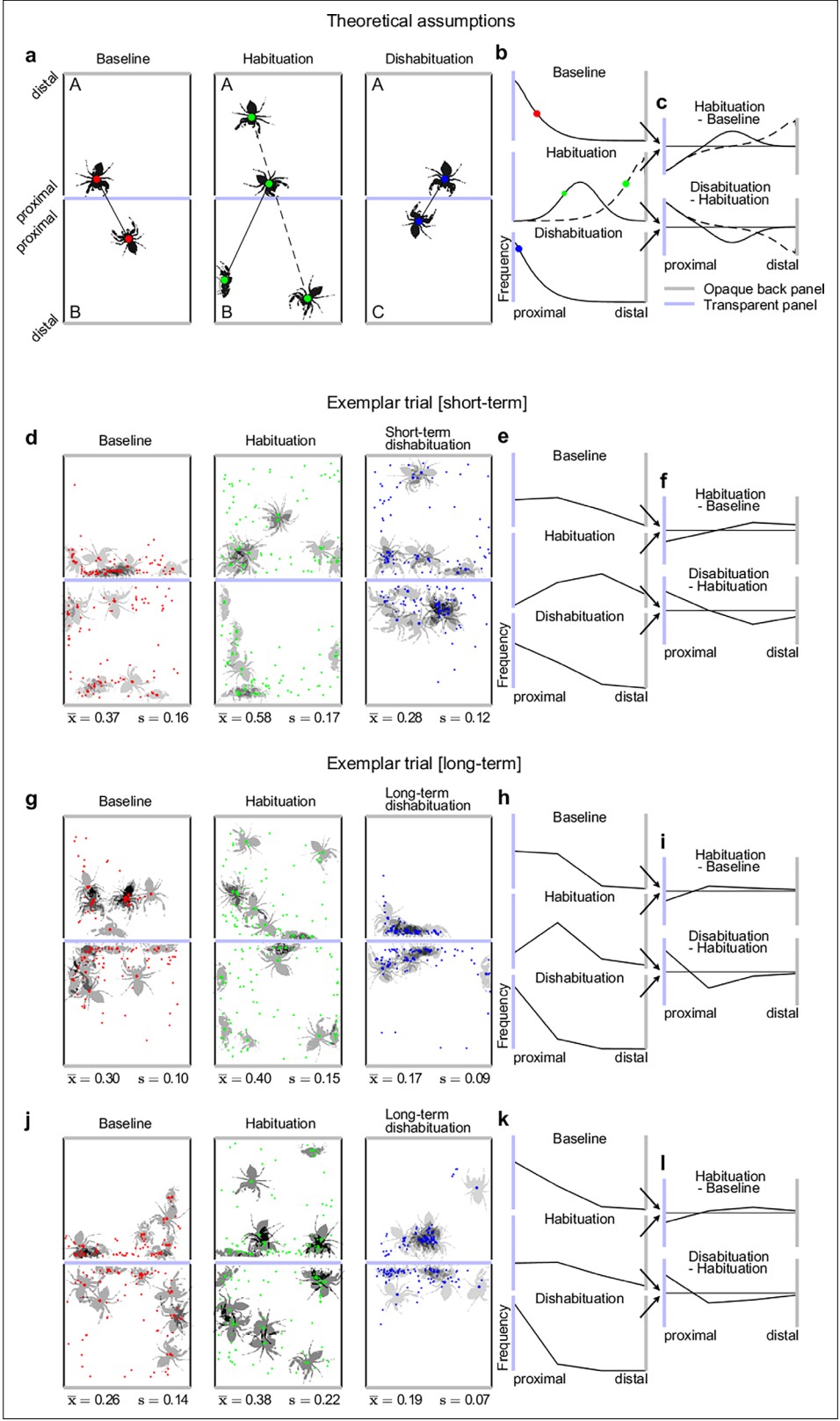

**Figure 1.** Theoretical assumptions and exemplar trials. (**a**-**c**) Predicted spider distances for *baseline* (red dots), *habituation* (green dots), and *dishabituation* comparisons (blue dots). Habituation can manifest either in equal inter-spider distances (solid line) as in the *baseline* comparison or in an increase of distances (dashed line). What is referred to as *baseline* in this context is the *dishabituation* trial of the previous comparison (see *Table 2*).

*Figure 1 continued on next page*

*Figure 1 continued*

Distance samples are predicted to fall into distributions as shown in b. Contrasts between *baseline*, *habituation*, and *dishabituation* comparisons would result in distributions as shown in c. (**d-f**) An exemplar trial consisting of *baseline*, *habituation,* and *dishabituation* comparisons from the first session of trials is shown. The *short-term dishabituation* comparison shows a decrease of inter-spider distances, indicating increasing interest in a different individual than the previously perceived one (*habituation* comparison). (**g-i**; **j-l**) Two exemplar trials from the third session of Experiment 2 are shown, where a presentation of an individual novel and unseen across the three experimental sessions triggered a great rebound in interest (i,l, 'Dishabituation - habituation'). (**d**, **g**, **j**) Note that in all upper quadrants, the same spider is used for baseline, habituation, and dishabituation comparisons, while in the lower quadrants, the baseline and habituation involve one individual, and a different (novel) spider is presented in the dishabituation comparison.

of model fitting the interaction of the predictors *distance* and *condition* (LRT: $\chi^2_{\Delta 3}$ = 63.66, p< 0.001; *Figure 2a*, exemplar trials: *Figure 1d–f*, *Video 1*, *Figure 2—videos 1; 2*; for model parameter estimation: *Appendix 1—figure 1*, *Appendix 1—table 1*): *dishabituation* trials (blue discs) showed a greater proportion of close-distance values than *habituation* trials (red discs), whereas *habituation* trials showed a greater proportion of far-distance values.

Furthermore, the strength of this effect varied across sessions. The interaction between *distance* and *condition* was significantly modulated by *session*, indicating that the dissociative effect of *condition* changed over the course of testing periods. This effect was strongest in Session 1 and weakened progressively, with the weakest modulation observed in Session 3 (LRT: $\chi^2_{\Delta 6}$ = 34.14, p< 0.001; *Figure 2a*, exemplar trials: *Figure 1d–f*, *Video 1*, *Figure 2—videos 1; 2*; for model parameter estimation: *Appendix 1—figure 1*, *Appendix 1—table 1*, *Dahl and Cheng, 2024*).

The systematic dissociation of distance values between *habituation* and *dishabituation* trials suggests that *P. regius* is capable of individual recognition. If spiders did not differentiate between conspecifics, we would not expect such a consistent divergence in distance patterns between conditions (*habituation* vs *dishabituation*). In addition, this dissociation diminished across sessions, indicating a progressive habituation effect towards the already encountered individuals. That is, in Session 1, the initial exposure to each individual elicited a pronounced dishabituation response. By Session 2, this dishabituation response was still present but attenuated. By Session 3, the dishabituation response had largely disappeared, indicating that spiders no longer differentiated between familiar and novel individuals (*Video 1*, *Figure 2—videos 1; 2*). While this progressive reduction is consistent with the formation and retrieval of individual-specific memories, alternative explanations, such as general fatigue due to extended testing periods, cannot be ruled out at this stage.

As a second step, we therefore examined whether *P. regius*'s decreasing interest across repeated sessions is driven by a general fatigue effect due to the prolonged testing procedure, or whether *P. regius* recognises the current individual after having encountered it at least once (by Session 2) or twice (by Session 3), and as a result, no longer dishabituates. Such recognition capability would support the presence of long-term memory representations in the individuation of conspecifics, given the extended retention interval spanning from minutes to hours.

To this end, we assessed whether the presentation of a completely novel individual - unseen during any of the three experimental sessions - would elicit a rebound in interest when introduced at the end of Session 3. We refer to this condition as *dishabituation [long-term]* trials, in contrast to the *dishabituation* trials conducted during Sessions 1–3, henceforth referred to as *dishabituation [short-term]* trials (see *Table 3*). Importantly, the labels 'short-term' and 'long-term' do not imply specific memory systems, but rather reflect the retention interval at which the respective *dishabituation* trials were administered.

**Table 1.** Pairwise comparisons.

| Trial | Pair 1 | Pair 2 |
| --- | --- | --- |
| 1 | A - B | C - D |
| 2 | A - C | B - D |
| 3 | B - C | A - D |

**Table 2.** Procedure of Experiment 1.

| Trial | Pair 1 | Pair 2 | | |
|---|---|---|---|---|
| 1 | A - B | C - D | | |
| 2 | A - B | C - D | {Habituation} | |
| 3 | A - C | B - D | | {Dishabituation} |
| 4 | A - C | B - D | {Habituation} | |
| 5 | B - C | A - D | | {Dishabituation} |
| 6 | B - C | A - D | {Habituation} | |
| … | | | | {Dishabituation} |
| 3 sessions | | | | |

If such rebound occurs, it would indicate that the observed habituation across sessions results from the recognition of repeatedly presented individuals, rather than from physical fatigue due to the prolonged testing procedure. In other words, such rebound would suggest a form of 'cognitive' fatigue – a diminished response elicited by the repeated re-encounter of familiar individuals – subserved by long-term memory formation, rather than a 'physical' fatigue effect. To test this, we re-ran the experiment in an additional 16 spiders, arranged into four groups and added a memory *dishabituation [long-term]* trial at the end of Session 3. The memory *dishabituation [long-term]* trials were generated by cross-combining individuals from two groups (group 1: A, B, C, D; group 2: E, F, G, H; *Table 3*) that had been run in parallel: At the end of Session 3, each spider was paired with a novel individual from the other group (e.g. A - E, B - G, C - F, D - H), resulting in previously unseen pairings.

First, we observed that spiders responded with renewed interest toward these unfamiliar individuals (*dishabituation [short-term]* trials), approaching them more closely than they had approached previously encountered ones (*habituation* trials). Thus, we replicated our earlier findings and found a dissociation between the factors *distance* and *condition* (LRT: $\chi^2_{\Delta 3}$ = 29.52, p < 0.001; *Figure 2b*, *Figure 2—video 3*; for model parameter estimation: *Appendix 1—figure 1*, *Appendix 1—table 2*). Specifically, *dishabituation [short-term]* trials (blue discs) showed a greater proportion of close-distance values, whereas *habituation trials* (red discs) showed a greater proportion of far-distance values. Most critically, we found that the *dishabituation [long-term]* trials at the end of Session 3 elicited a rebound in interest that clearly exceeded the rebound in the *dishabituation [short-term]* trials of the same session. This was reflected in a significant interaction between *condition* (i.e. *dishabituation [short-term]* vs *dishabituation [long-term]*) and *distance* (F(3,127) = 3.91, sum sq.=0.92, mean sq.=0.31; p< 0.01, *Figure 2b*: right subfigure, white diamonds; exemplar trials: *Figure 1g–i and j–l*, *Videos 2 and 3*, *Figure 2—video 4*, *Dahl and Cheng, 2024*). This finding suggests that the reduction in effect size across sessions was not due to general fatigue, but rather reflects a long-term memory for previously encountered individuals: That is, when presented with a truly novel conspecific, the spiders' interest rebounded to an unprecedented level.

Our findings show, first, that *P. regius* can recognise individuals to which it was exposed to for a short period of 7 min and that reoccurred after a visual separation period of 3 min. Second, *P. regius* exhibited long-term habituation, that is a sustained reduction in interest when encountering the same individuals again 1 or 2 hr later. This result pattern is only possible if spiders remember the encountered individuals across sessions, suggesting the formation and retention of individual-specific memory representations. Third, despite this long-term habituation, *P. regius* showed an unprecedented rebound in interest when confronted with an entirely novel individual, ruling out a physical fatigue effect in favour of a 'cognitive' fatigue based on long-term memory. For these reasons, our results are the first evidence that *P. regius*, a non-social arthropod species, possesses long-term memory that allows it to individuate conspecifics and recognise novel individuals.

## Discussion

Before addressing the evolutionary implications of these findings, it is important to consider a key methodological question: Do the observed behavioural changes reflect true recognition memory, or

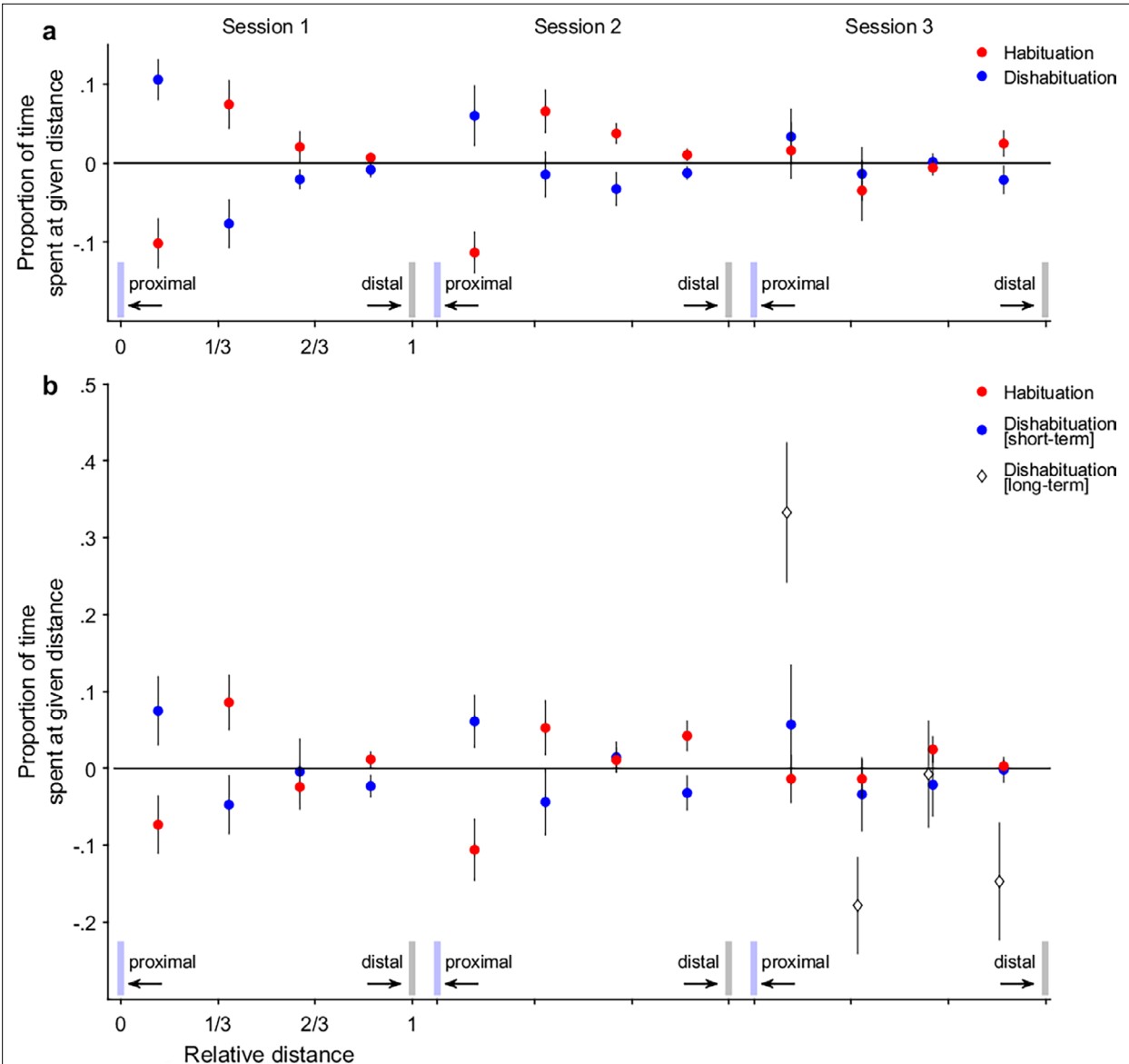

**Figure 2.** The relative change in distance between pairs of individuals, upon being confronted with the same individual as in the preceding trial (*habituation* trial; red discs) or a different individual from the individual in the preceding trial (*dishabituation* trial; blue discs). Each panel refers to an experiment (panel a. for Experiment 1; panel b. for Experiment 2), consisting of three sessions of trials. The dependent data is shown as the proportion of time spent at a given distance binned into 4 equally spaced bins. The x-axis labels refer to the proportional distances from the transparent acrylic sheet, ranging from 'proximal' to 'distal'; the y-axis refers to the proportion of time spent at a given distance, that is the relative number of samples that fall into a given bin. Discs show the mean proportion across all individuals (i.e. 20 for Experiment 1; 16 for Experiment 2). The whiskers indicate the standard errors of the mean. White diamonds in the lower right subfigure b show the *long-term dishabituation* trials. Light blue bars indicate the side of the transparent acrylic sheet (proximal); grey bars indicate the back wall of the container (distal).

The online version of this article includes the following video(s) for figure 2:

**Figure 2—video 1.** Exemplar *dishabituation [short-term]* trial from Experiment 1.
https://elifesciences.org/articles/97146/figures#fig2video1

**Figure 2—video 2.** Exemplar *dishabituation [short-term]* trial from Experiment 1.
https://elifesciences.org/articles/97146/figures#fig2video2

**Figure 2—video 3.** Exemplar *dishabituation [short-term]* trial from Experiment 2.
https://elifesciences.org/articles/97146/figures#fig2video3

**Figure 2—video 4.** Exemplar *dishabituation [long-term]* trial from Experiment 2.
https://elifesciences.org/articles/97146/figures#fig2video4

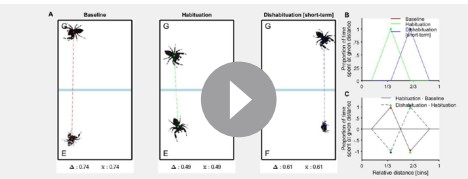

**Video 1.** Exemplar *dishabituation [short-term]* trial from Experiment 1. a. Baseline, habituation, and dishabituation trials are shown for a given individual (top half of the black box) with varying partner according to condition (lower half of the black box), i.e. baseline and habituation trials require an exposure to an identical partner; dishabituation trials to a different partner. Distances are indicated by a coloured dashed line (red for baseline, green for habituation, blue for dishabituation trials). The light blue lines dividing the black boxes illustrate the approximate placement of the transparent acrylic sheets. The black boxes indicate the approximate location of the walls of the containers. The exact location of walls and transparent front panels might slightly vary. The $\Delta$-values show the current relative distance between individuals for a given condition and $\bar{x}$-values the mean relative distance up to the current sample. The maximal distance, i.e., when both spiders are in diagonally opposite corners, is 1, the minimal distance is 0. b. The distribution of relative distances of data samples up to the current point in time is shown as proportion of time spent at a given distance according to the trial types (baseline, habituation, dishabituation). The discs indicate the bin to which the current distance values are assigned to, and, hence, dynamically change their location as the spider moves. c. Trial types are shown as subtraction from each other, such that habituation trials are contrasted with baseline trials (black line), and dishabituation trials with habituation trials (dashed black line). Similarly to b., the discs indicate to which bin the current sample is assigned. The vertical positioning of the discs indicates by their colours which trial type is more frequent at a given point, e.g., a blue disc located above a green disc indicates that for the given bin the dishabituation trial (blue disc) showed more values falling into that bin than the habituation trial (green disc) ($\equiv$ positive value); a green disc located above a blue disc indicates that for the given bin the habituation trial (green disc) showed more values falling into that bin than the dishabituation trial (green disc) ($\equiv$ negative value).

https://elifesciences.org/articles/97146/figures#video1

could they result from simpler mechanisms such as transient familiarity or physical fatigue? To distinguish these possibilities, we employed a stringent habituation–dishabituation paradigm (*Fantz, 1961*), widely used in animal (*Dahl et al., 2007*) and infant cognition (*Kavsek and Bornstein, 2010*) to isolate memory-based recognition from short-term novelty effects. Unlike paradigms with static images or single exposures, our approach required subjects to recognise a dynamic, interacting conspecific across repeated encounters, making low-level novelty detection or general desensitisation unlikely explanations. In our design, evidence for individual recognition comes from both within-session and across-session comparisons: Within sessions, *dishabituation [short-term]* trials consistently elicited closer proximity-seeking behaviour than *habituation* trials, indicating discrimination between familiar and novel individuals. Across sessions, this effect showed a structured decline: strong in Session 1, attenuated in Session 2, and absent in Session 3. This temporal pattern is inconsistent with a general desensitisation, which would reduce proximity-seeking behaviour across all trial types regardless of stimulus identity. Instead, our results support the formation of identity-specific memory representations. Further support comes from the *long-term dishabituation* trials at the end of Session 3, where introducing a novel individual elicited the strongest rebound in interest. If prior responses reflected transient familiarity or physical fatigue, such a pronounced effect would not be expected. This indicates that spiders encoded previous individuals as distinct representations and recognised novelty at the level of individual identity.

Our results provide clear evidence of memory-based individual recognition in *P. regius*. If a spider does not dishabituate when re-encountering an individual, this indicates that a memory representation for that individual has been formed and retrieved. Conversely, if a marked dishabituation response occurs upon introduction of a novel individual, it implies that the current experience does not match any existing memory representation, prompting renewed interest. The clear dissociation between *habituation* and *dishabituation* trials within sessions, the progressive reduction in dishabituation responses across sessions, and the pronounced rebound upon introduction of a truly novel individual together indicate the formation and retrieval of individual-specific memories, rather than transient familiarity or physical fatigue.

Recognising members of one's own species is a crucial cognitive ability that underpins various adaptive behaviours. Individual recognition allows animals to distinguish between friend and foe, to identify a mating partner, its offspring, or a kin member. Individual recognition is achieved via the

**Table 3.** Pairwise comparisons as *habituation* and *dishabituation* trials.

|       | Group 1 | | Group 2 | | | |
|-------|---------|--------|---------|--------|-------------------|-------------------|
| Trial | Pair 1  | Pair 2 | Pair 1  | Pair 2 | | |
| 1     | A - B   | C - D  | E-F     | G-H    | | |
| 2     | A - B   | C - D  | E-F     | G-H    | {Habituation}     | |
| 3     | A - C   | B - D  | E-G     | F-H    |                   | {Dishabituation}  |
| 4     | A - C   | B - D  | E-G     | F-H    | {Habituation}     | |
| 5     | B - C   | A - D  | F-G     | E-H    |                   | {Dishabituation}  |
| 6     | B - C   | A - D  | F-G     | E-H    | {Habituation}     | |
| …     |         |        |         |        | | |
| 3 sessions | |     |         |        | | |
| 7     | A-E     | B-G    | C-F     | D-H    |                   | {Dishabituation}  |

production of individually distinct features (e.g. visual) or signals (e.g. acoustic) by the sender, and the ability of the receiver to extract and process these features and signals (*Yorzinski, 2017*). In social species, individual recognition bears particular significance in contexts such as territoriality, aggressive competition, and parental care (*Tibbetts and Dale, 2007*). It is precisely because jumping spiders, including *P. regius*, are generally considered non-social, solitary, and aggressive towards conspecifics that the presence of individual recognition is unexpected. This raises the question about the biological relevance of individual recognition in *P. regius*: One of the few social interactions in the life of a jumping spider occurs during mating, encompassing a highly structured visual courtship display composed of coordinated body movements and distinct morphological features. It is believed that the colouration of the appendages (the chelicerae), along with the facial hair patterns, serves as important visual features for species and sex identification in jumping spiders and may also signal individual quality during mate assessment (*Zhou et al., 2021*; *Hill et al., 2006*). Hence, the colouration of body parts (sender) and the ability to discriminate colours (receiver) appear sufficient for sexual selection, such as species and sex identification or mate assessment, without necessarily supporting individual recognition (*Lim et al., 2007*; *Zhou et al., 2021*). Similarly, in aggressive interactions, such as territorial disputes, fighting ability is largely associated with the size and colouration of the chelicerae (*Lim et al., 2007*; *Tedore and Johnsen, 2012*). In line with this, research has shown that the demands of mating and aggression in jumping spiders are met by cue recognition - by definition a basic-level classification - allowing classification of species, sex, or general rival status. For example, jumping spiders have been shown to use a combination of visual and chemical (pheromonal) cues to differentiate both species and sex, enabling them to identify potential mates or competitors (*Tedore and Johnsen, 2013*). Some species recognise rivals by relying on distinctive colour signals, such as a red facial patch, which serves as a key visual cue in aggressive

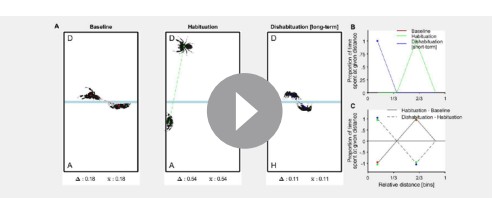

**Video 2.** Exemplar *dishabituation [long-term]* trial from Experiment 2, focusing on Individual D. Baseline, habituation, and dishabituation trials are shown following the same conventions (baseline = red, habituation = green, dishabituation = blue). Dynamic distance distributions and proportional occupancy of bins are displayed as in Video Figures and Tables.
https://elifesciences.org/articles/97146/figures#video2

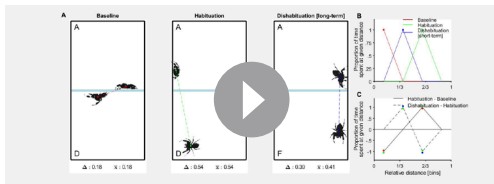

**Video 3.** Exemplar *dishabituation [long-term]* trial from Experiment 2, same trial as *Video 2* but focusing on Individual A. Baseline, habituation, and dishabituation trials are shown following the same conventions (baseline = red, habituation = green, dishabituation = blue). Dynamic distance distributions and proportional occupancy of bins are displayed as in Video Figures and Tables.
https://elifesciences.org/articles/97146/figures#video3

interactions (*Cross et al., 2020*). In each case, it is basic-level category recognition rather than individual identity that guides behaviour, and thus subordinate-level discrimination appears unnecessary for these core ecological functions. As such, territoriality and aggressive behaviour appear insufficient as ultimate explanations (*Tinbergen, 1963*) for the evolution of individual recognition in *P. regius*. Instead, decisions in these contexts can be based on general physical features rather than identity-specific information.

In addition, most jumping spiders exhibited limited parental care (*Gardner, 1965*), protecting the nest through the spiderlings' first molt. A notable exception is *Toxeus magnus*, which provides a nutrient-rich, milk-like substance to its offspring, a behaviour functionally and behaviourally analogous to lactation in mammals (*Chen et al., 2018*). Despite these examples, there is little evidence that memory-based recognition of individual offspring is required to support parental care in salticids. In more specific contexts, however, some jumping spiders have been shown to discriminate their own eggsacs from those of conspecifics, presumably to avoid filial cannibalism (*Clark and Jackson, 1994a*), or to distinguish their own silk draglines from those of other conspecifics, representing a form of self-recognition (*Clark and Jackson, 1994b*). Both forms of recognition, however, are context-dependent and do not extend to memory-based identification of conspecifics across repeated encounters. Hence, although jumping spiders demonstrate a range of basic-level recognition abilities adapted to specific ecological challenges, these abilities, however, do not extend to the subordinate-level individual recognition demonstrated in the present study. In other words, even in this context, there is no clear evidence that memory-based individual recognition solves a critical survival problem.

Nevertheless, it is worth considering that individual recognition could provide adaptive benefits in certain social contexts, even in taxa where repeated encounters are typically rare. For instance, the so-called 'dear enemy effect', a phenomenon observed in territorial animals, involves reduced aggression towards familiar neighbours while maintaining heightened vigilance against unfamiliar individuals (*Temeles, 1994*; *Bee and Gerhardt, 2002*). This form of recognition can minimise the costs of unnecessary conflict and enhance social stability. While the natural history of *P. regius* suggests rather infrequent repeated encounters with the same conspecific, the current findings raise the intriguing possibility that cognitive capacities for identity recognition facilitate the emergence of 'dear enemy' relationships if territorial dynamics arise. Future experiments could directly test whether the 'dear enemy effect' can be experimentally elicited in jumping spiders.

Taken together, the generally solitary nature of *P. regius* offers little ecological pressure to favour individual recognition. According to the 'minimum needs hypothesis' (*Wiley, 2013*; *Gherardi et al., 2012*), selection will only favour specific recognition abilities when basic-level classification (*Rosch et al., 1976*; e.g. species, sex, or quality assessment via colouration or size cues) no longer suffices to solve ecologically relevant problems. In the case of *P. regius*, basic discrimination, such as chelicerae size or colouration, may be sufficient in most known contexts of social interaction. Moreover, the neural implementation of subordinate-level classification (*Rosch et al., 1976*), such as identity recognition, requires more detailed and exhaustive processing than basic-level categorisation. This type of fine-grained discrimination is typically associated with specialised neural structures and increased cognitive and neural resource requirements (*Weiner and Grill-Spector, 2015*). Given these costs and the limited ecological need for identity recognition in *P. regius*, it is more plausible to assume that this ability arises not through direct selection, but through pleiotropy, as a byproduct of general cognitive architecture evolved for other purposes - a view consistent with the 'general learning hypothesis' (*Wiley, 2013*). Jumping spiders are known for complex foraging and navigation strategies (*Jakob et al., 2007*; *Jakob et al., 2011*; *Skow and Jakob, 2006*), requiring flexible learning and high behavioural adaptability. These domain-general cognitive capabilities may enable recognition of conspecifics with a degree of detail and specificity that exceeds their minimum recognition needs (*Yorzinski, 2017*).

In contrast to *social* animal species, including eusocial arthropod species (*Tibbetts, 2002*; *Sheehan and Tibbetts, 2011*; *Dreier et al., 2007*), where individual recognition is often maintained by direct selective advantage, we cannot conclusively identify a survival benefit for individual recognition in *P. regius*. Instead, we put forward the idea that individual recognition in *P. regius* emerges as a pleiotropic consequence, namely a byproduct of an already fairly sophisticated, broader learning capability. Critically, individual recognition relies on recognition memory, a form of long-term memory, in which a previously encountered event or entity, here an individual, is stored as a representation and reactivated

upon re-encounter (*Norman and O'Reilly, 2003*). This memory mechanism may also underlie complex behaviours observed in other salticids, such as when *Portia fimbriata* navigates towards prey after an initial visual scan of the environment - even without visual input during execution (*Tarsitano and Jackson, 1997*) - or when *Menemerus semilimbatus* recognises biological motion, demonstrating sophisticated temporal integration and dynamic cue recognition (*De Agrò et al., 2021*).

Thus, our study challenges the notion of spiders being stimulus-response driven automata, by not only contributing to an increasing body of evidence that spiders and salticids in particular produce a wide spectrum of intelligent behaviour (*Jackson and Cross, 2011*), but by pinpointing the presence of two fundamentally important mechanisms for any higher cognitive processing: flexible learning and recognition memory. The key building blocks of these mechanisms are representations, mental images of external entities, that are not present to the sense organs, allowing more elaborate information processing, such as in complex decision making and goal-directed behaviour. The existence of which in arthropods in general and spiders in particular triggers rethinking of miniature brain cognition (*Jackson and Cross, 2011*).

## Materials and methods

### Subjects

Our subjects were 36 jumping spiders (*Phidippus regius*), kept individually in enclosures (7x7 x 12 cm) at room temperature (21–25 °C) and supplied with a moist water pad, exchanged every other day, and two small-sized cockroaches (*Shelfordella lateralis*) per week. All spiders were adult laboratory-bred and had no direct encounters with conspecifics during adulthood. Behavioural enrichment (*Carducci and Jakob, 2000*) was provided by means of climbing and nesting structures (i.e. natural wood branch) and by interaction with human caretakers and experimenters during handling and maintenance procedures. In Experiment 1, spiders were assigned to five experimental groups, three of which contained females, two of which males; in Experiment 2, spiders were assigned to four experimental groups, that is two groups per sex.

### Apparatus

In the following, we describe how pairs of spiders were brought into direct visual contact under controlled conditions and in a manner that allows us to reassign individuals easily and without interruption to form novel pairings. To this end, we built a cubical experimental arena of 60x46 x 65cm (L x W x H), consisting of white polypropylene plastic panels, mounted in a frame of T-slotted aluminium profiles (20 Series; Misumi Group Inc, Bunkyo City, Tokyo, Japan). Two LED light sources (Mettle SL400, 45 W, 2100lm, 350x250 mm surface area, Mettle Photographic Equipment Corporation, Changzhou, China) were placed outside the cubicle at 25 cm distance from the side panels of the cubicle, illuminating the inside of the cubicle uniformly. We also mounted two FLIR 1.3MP, Mono Blackfly USB3 cameras with a 1/2" CMOS sensor (BFS-U3-13Y3M-C, FLIR Integrated Imaging Solutions, Inc, 12051 Riverside Way, Richmond, BC, Canada) equipped with 8 mm UC Series lenses from Edmund Optics (Stock #33–307, Edmund Optics, Barrington, New Jersey, USA) on T-slotted aluminium profiles, facing downwards onto the arena surface at a distance of 60 cm. For each spider, we 3D-printed a white container with outer dimensions (L x W x H) of 7x7 x 5cm and inner dimensions of 6.3x6.3 x 4.5cm. The upper side of the container and one of the four side walls were made of a transparent.5-mm-thick acrylic sheet. While the acrylic sheet on the upper side of the container was screwed onto the side walls of the container, the acrylic sheet on one of the sides of the container can be lifted up to open the container, allowing easier transfer of the spider from the home enclosure.

### Procedure

In Experiment 1, each group consisted of four same-sex spiders, with each spider being placed inside a container prior to the experiment. We allowed the spiders sufficient time (10–15 min) to acclimatise to the new environment. During the experiment, the spiders remained in their own containers. With the start of the experiment, we then placed the containers of the four spiders such that the transparent side walls of two containers were facing each other, resulting in two pairs of spiders with direct visual contact to each other. During the arrangement of the containers and before each new trial, visual contact was blocked using an occluder placed between the transparent side walls of the

containers. Each trial was initiated by removing this occluder, allowing visual contact. For simplicity, let the four individuals be symbolised by the letters 'A', 'B', 'C', and 'D': An arrangement of trials where each individual is opposed to each other individual is described in *Table 1*.

To tease apart, whether or not *P. regius* was capable of visually discriminating other individuals, two types of trials were required: (a) a *habituation* trial, where the same individual was presented in the trial preceding the current trial (e.g. trial 1: A - B, trial 2: A - B), and (b) a *dishabituation* trial, where a different individual was presented in the trial preceding the current trial (e.g. trial 1: A - B, trial 2: A - C). Thus, each *dishabituation* trial followed a *habituation* trial, forming alternating *habituation* and *dishabituation* phases, respectively, as shown in *Table 2*.

In detail, a trial, for example A - B (and in parallel C - D), lasted for 7 min allowing the spiders to visually inspect each other, before isolating the spiders visually for 3 min with a non-transparent white occluder, fully covering the transparent side wall. Following the occluder phase, another 7 min exposure phase was initiated, which consisted of either the same individual (*habituation* trial) or another individual (*dishabituation* trial) than in the preceding trial. During each trial, the distance between individuals was measured at 10 Hz temporal resolution and used as an indicator of 'interest': shorter distances signalled greater interest in the other individual, while longer distances indicated reduced interest.

This experimental design was specifically structured to assess identity recognition memory in *P. regius*. With the repetition of trials, *habituation,* and *dishabituation* phases, we can assess whether a currently visually encountered individual (*habituation*) elicits a differential behavioural response to a novel individual (*dishabituation*). We predict a dissociation of distances between *habituation* and *dishabituation* trials. This habituation-dishabituation paradigm is widely used in developmental (*Kavsek and Bornstein, 2010*) and animal cognition studies (*Dahl et al., 2007*) to evaluate recognition memory in non-verbal subjects.

With the outlined procedure (*Table 2*), we can form sequences of trial phases, where each first of two trials is a *habituation* trial, and every second is a *dishabituation* trial. Thus, a *habitation* trial consists of a repeated pairing, such as A - B followed by A - B, while a *dishabituation* trial consists of a novel pairing, such as A - B followed by A - C. As a result, each trial within a trial-phase subserves both the *habituation* as well as the *dishabituation* trial. In this manner, we created a trial list, containing 12 trials in total, six of which result in *habituation* trials and six of which result in *dishabituation* trials (*Table 2*). This session of trials was repeated twice, resulting in a total of 36 trials per experiment. Each experiment lasted 180 min, where each trial contained 7 min exposure and 3 min visual separation. Each group of spiders was subjected to this protocol.

Two amendments were made in Experiment 2: (a) we ran two groups of four individuals in parallel, and (b) we introduced additional *cross-group* trials were introduced at the end of Session 3. This resulted in a modified procedure described in *Table 3*.

## Data logging and analysis

Camera control and image acquisition were done using Matlab (Mathworks, Natick, Massachusetts, USA) and the image acquisition and processing toolboxes. The frame rate was set to 10 Hz. Cameras were placed perpendicular to the xy-plane at a distance of about 60 cm from the ground. The lens aperture was set to f/4, allowing a sufficient depth of field. Analysis was done with Matlab (Mathworks, Natick, Massachusetts, USA). We pre-processed the video recordings by segmenting the spider body from the background in each frame using functions for image intensity adjustment, image enhancement, image binarization, and image properties measurement to extract the largest available 'region', the spider body, and its centroid. For each trial, we approximated the distance between the individuals in the xy-plane as a function of time, using the Euclidean distance weight function based on the centroid coordinates of the two individuals. We then pooled the distance values of each trial into 4 equally sized and non-overlapping bins (bin centers [mm]: [20, 60, 100, 140]; bin size 40 mm; maximal distance ≈160 mm) and calculated the proportion of time spent at a given distance. Each bin was normalised by the total number of events. Differences between proportions were then calculated for every trial comparison according to *Tables 2 and 3*: For instance, the proportions of time spent

at a given distance for individual A in trial 1 were subtracted from the proportions of time spent at a given distance for individual A in trial 2, resulting in an assessment for the relative rebound of interest following a repetition of exposure to the same spider B (habituation). Subsequently, the proportions of time spent at a given distance for individual A in trial 2 were subtracted from the proportions of time spent at a given distance for individual A in trial 3, resulting in an assessment for the relative rebound of interest following changes in spider identity (dishabituation). We used linear mixed-effects models, where the differences in proportions served as the dependent variable. We fitted two separate models for each experiment (*Full model 1* and *2*), and followed a commonly accepted model fitting procedure (*Dobson and Barnett, 2018*): To fully account for the dependent variable, we fitted three predictor variables: (1) The bin number ([1–4]), reflecting a discretised distance measure and henceforth referred to as factor *distance* ([1–4]), (2) the *session* of comparisons (*Anderson and McShea, 2001*; *Bee and Gerhardt, 2002*; *Bonatti et al., 2002*), as outlined in the Procedure above (*Tables 2 and 3*), and (3) the *condition*, referring to whether the given comparison was a *habituation* or *dishabituation* comparison. We also fitted all two-way interactions between the three main predictors: *distance:session*, *distance:condition*, and *session:condition*, as well as the three-way interaction *distance:session:condition*. Of particular interest are the two-way interaction between the factors *distance* and *condition*, since we predict a modulation of *distance* values by *condition* as a function of *distance*, and the three-way interaction between the factors *distance*, *condition,* and *session*, since we predict a modulation of *condition* as a function of *distance* which becomes weaker over time and repetitions, i.e. *session*. We further defined *sex* of the subjects and *subject* as random factors in all models. We fitted a linear mixed-effects model (fitlme function in Matlab) with normal error structure and identity link function to our data set. We then created a null model for each corresponding full model, which consisted of the similar structure as the full model, however leaving only *distance* as fixed effect, while preserving all random effects. Using the likelihood ratio test (LRT), we compared the null models with the corresponding full models. Assuming a significant improvement for the full model over the null model, the non-significant interaction terms were removed from the full model, reaching a model containing only significant interaction terms and both significant and non-significant main effects (*Hector et al., 2010*; *Forstmeier and Schielzeth, 2011*), henceforth referred to as the final model. Evaluation of fixed effect were on the basis of the final models and are referred to as the *Final model 1* (Experiment 1), and *Final model 2* (Experiment 2; *Appendix 1—figure 1*, *Appendix 1—tables 1 and 2*; *Dahl and Cheng, 2024*). This procedure resulted in the following models (Wilkinson notation):

Final models 1 and 2:

$$'\text{Response} \sim 1 + \text{Distance} + \text{Session} + \text{Condition} + \text{Distance} : \text{Condition} + ...$$
$$\text{Distance} : \text{Session} : \text{Condition} + (1 \mid \text{Sex}) + (1 \mid \text{Subject})'$$

An additional analysis of variance was performed comparing the *dishabituation [long-term]* trials at the end of Session 3 with the *dishabituation [short-term]* trials from Session 3 (*Table 3*) as a function of *distance*. No statistical methods were used to predetermine sample size. The experiments were not randomized. The investigators were not blinded to allocation during experiments and outcome assessment.

## Acknowledgements

We are grateful for the financial support by the Swiss National Science Foundation (PZ00P3_154741), the Startup-funding of Taipei Medical University (TMU108-AE1-B33) and the Taiwan Ministry of Science and Technology research grants (110–2311-B-038–002, 112–2410 H-038-027) awarded to CDD. We thank Guillaume Dezecache, Niall W Duncan, Olivier Pascalis, Tzu-Yu Hsu, Werner Müller and Timothy J Lane for suggestions and comments on the manuscript.

# Additional information

### Funding

| Funder | Grant reference number | Author |
|---|---|---|
| Taipei Medical University | TMU108-AE1-B33 | Christoph D Dahl |
| National Science and Technology Council | 110-2311-B-038-002 | Christoph D Dahl |
| National Science and Technology Council | 112-2410-H-038-027 | Christoph D Dahl |
| Swiss National Science Foundation | PZ00P3_154741 | Christoph D Dahl |

The funders had no role in study design, data collection and interpretation, or the decision to submit the work for publication.

### Author contributions

Christoph D Dahl, Conceptualization, Resources, Data curation, Software, Formal analysis, Supervision, Funding acquisition, Validation, Investigation, Visualization, Methodology, Writing – original draft, Project administration, Writing – review and editing; Yaling Cheng, Conceptualization, Resources, Investigation, Methodology, Project administration

### Author ORCIDs

Christoph D Dahl ⓘ https://orcid.org/0000-0002-4296-1526
Yaling Cheng ⓘ https://orcid.org/0009-0002-5198-5797

### Ethics

According to Taiwans Animal Protection Act, issued by the Council of Agriculture (Executive Yuan), experiments on invertebrates are allowed to be conducted without any special permission in Taiwan.

Reviewer #1 (Public review): https://doi.org/10.7554/eLife.97146.3.sa1
Reviewer #3 (Public review): https://doi.org/10.7554/eLife.97146.3.sa2
Author response https://doi.org/10.7554/eLife.97146.3.sa3

---

# Additional files

### Supplementary files
MDAR checklist

### Data availability

Supplementary information is available for this paper. Codes and materials are available (https://osf.io/gpnct/).

The following dataset was generated:

| Author(s) | Year | Dataset title | Dataset URL | Database and Identifier |
|---|---|---|---|---|
| Dahl CD, Cheng Y | 2023 | Individual recognition memory in a jumping spider | https://doi.org/10.17605/OSF.IO/GPNCT | Open Science Framework, 10.17605/OSF.IO/GPNCT |

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

## Appendix 1

## Model parameter estimations for Experiments 1 and 2

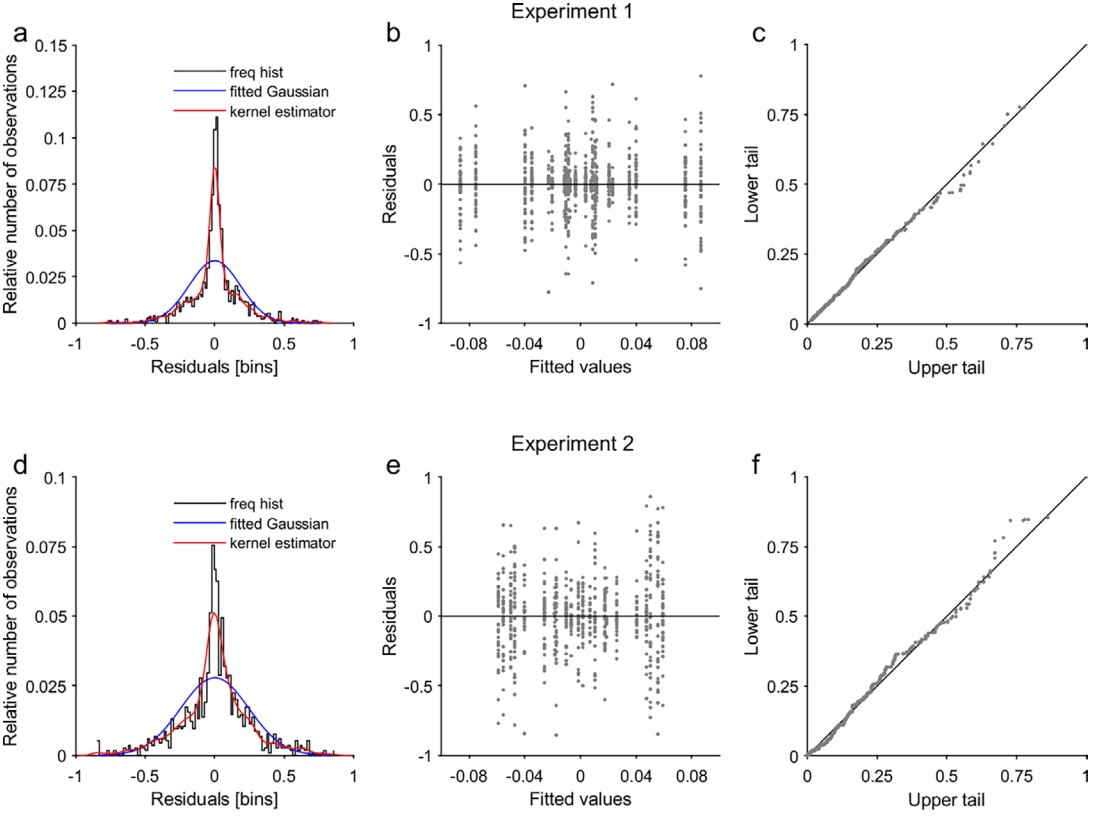

**Appendix 1—figure 1.** Residuals of the linear mixed-effects model. a, d. Residuals of linear mixed-effects model 1 (**a**) and 2 (**d**). The histograms bin the residuals for each model into 100 equally-spaced containers (x-axis) and return the number of elements (y-axis) in each container. The blue lines indicate a Gaussian fit; the red lines show a kernel distribution estimate. b, e. show the residuals (y-axis) plotted against the fitted values (x-axis) for model 1 (**b**) and model 2 (**e**). c, f. Residuals of lower and upper tails are plotted against each other, showing an equal distribution in both model 1 (**c**) and model 2 (**f**).

**Appendix 1—table 1.** Results of the model investigating pairwise subtracted frequency distribution of distance values.

The table contains parameter estimates for the final model 1 based on the fixed factors 'distance', 'session', 'condition', 'distance:condition', 'distance:session:condition', as well as the random factors 'sex' and 'subject'.

| | | Estimate | SE | t-stat | DF | p-value | CI (95%) [lower,upper] |
|---|---|---|---|---|---|---|---|
| | Intercept | –0.001 | 0.01 | –0.001 | 1424 | 1 | [–0.01; 0.01] |
| | Distance 1 | 0.001 | 0.01 | 0.001 | 1424 | 1 | [–0.02; 0.02] |
| | Distance 2 | 0.001 | 0.01 | 0.001 | 1424 | 1 | [–0.02; 0.02] |
| | Distance 3 | –0.001 | 0.01 | –0.001 | 1424 | 1 | [–0.02; 0.02] |
| Distance | (against *Distance* 4) | | | | | | |

*Appendix 1—table 1 Continued on next page*

*Appendix 1—table 1 Continued*

| | | Estimate | SE | t-stat | DF | p-value | CI (95%) [lower,upper] |
|---|---|---|---|---|---|---|---|
| | Session 1 | 0.001 | 0.01 | 0.001 | 1424 | 1 | [–0.01; 0.01] |
| | Session 2 | 0.001 | 0.01 | 0.001 | 1424 | 1 | [–0.01; 0.01] |
| Session | (against *Session* 3) | | | | | | |
| | Condition 1 | 0.001 | 0.01 | 0.001 | 1424 | 1 | [–0.01; 0.01] |
| Condition | (against *Condition* 2) | | | | | | |
| | Distance 1: Condition 1 | –0.07 | 0.01 | –7.90 | 1424 | 0.001 | [–0.08; –0.05] |
| | Distance 2: Condition 1 | 0.03 | 0.01 | 4.16 | 1424 | 0.001 | [0.02; 0.05] |
| | Distance 3: Condition 1 | 0.02 | 0.01 | 2.06 | 1424 | 0.05 | [0.01; 0.03] |
| Distance x Condition | (against *Distance* 4 and *Condition* 2) | | | | | | |
| | Distance 1: Session 1: Condition 1 | –0.04 | 0.01 | –3.14 | 1424 | 0.01 | [–0.06; –0.01] |
| | Distance 2: Session 1: Condition 1 | 0.04 | 0.01 | 3.41 | 1424 | 0.001 | [0.02; 0.06] |
| | Distance 3: Session 1: Condition 1 | 0.01 | 0.01 | 0.27 | 1424 | 0.78 | [–0.02; 0.02] |
| | Distance 1: Session 2: Condition 1 | –0.02 | 0.01 | –1.71 | 1424 | 0.08 | [–0.04; 0.01] |
| | Distance 2: Session 2: Condition 1 | 0.01 | 0.01 | 0.42 | 1424 | 0.67 | [–0.02; 0.03] |
| | Distance 3: Session 2: Condition 1 | 0.02 | 0.01 | 1.50 | 1424 | 0.13 | [–0.01; 0.04] |
| Distance x Session x Condition | (against *Distance* 4, *Session* 3 and *Condition* 2) | | | | | | |

**Appendix 1—table 2.** Results of the model investigating pairwise subtracted frequency distribution of distance values.

The table contains parameter estimates for the final model 2 based on the fixed factors 'distance', 'session', 'condition', 'distance:condition', 'distance:session:condition', as well as the random factors 'sex' and 'subject'.

| | Estimate | SE | t-stat | DF | p-value | CI (95%) [lower, upper] |
|---|---|---|---|---|---|---|
| Intercept | 0.01 | 0.01 | 0.01 | 1136 | 1 | [–0.01; 0.01] |

*Appendix 1—table 2 Continued on next page*

*Appendix 1—table 2 Continued*

| | | Estimate | SE | t-stat | DF | p-value | CI (95%) [lower, upper] |
|---|---|---|---|---|---|---|---|
| | Distance 1 | –0.01 | 0.01 | –0.01 | 1136 | 1 | [–0.02; 0.02] |
| | Distance 2 | 0.01 | 0.01 | 0.01 | 1136 | 1 | [–0.02; 0.02] |
| | Distance 3 | 0.01 | 0.01 | 0.01 | 1136 | 1 | [–0.02; 0.02] |
| Distance | (against *Distance* 4) | | | | | | |
| | Session 1 | 0.01 | 0.01 | 0.01 | 1136 | 1 | [–0.02; 0.02] |
| | Session 2 | –0.01 | 0.01 | –0.01 | 1136 | 1 | [–0.02; 0.02] |
| Session | (against *Session* 3) | | | | | | |
| | Condition 1 | –0.01 | 0.01 | –0.01 | 1136 | 1 | [–0.01; 0.01] |
| Condition | (against *Condition* 2) | | | | | | |
| | Distance 1: Condition 1 | –0.06 | 0.01 | –5.14 | 1136 | 0.001 | [–0.09; –0.04] |
| | Distance 2: Condition 1 | 0.04 | 0.01 | 3.32 | 1136 | 0.001 | [0.02; 0.07] |
| | Distance 3: Condition 1 | 0.01 | 0.01 | 0.30 | 1136 | 0.77 | [–0.02; 0.03] |
| Distance x Condition | (against *Distance* 4 and *Condition* 2) | | | | | | |
| | Distance 1: Session 1: Condition 1 | –0.01 | 0.02 | –0.55 | 1136 | 0.58 | [–0.04; 0.02] |
| | Distance 2: Session 1: Condition 1 | 0.02 | 0.02 | 1.41 | 1136 | 0.16 | [–0.01; 0.06] |
| | Distance 3: Session 1: Condition 1 | –0.01 | 0.02 | –0.77 | 1136 | 0.44 | [–0.05; 0.02] |
| | Distance 1: Session 2: Condition 1 | –0.02 | 0.02 | –1.09 | 1136 | 0.28 | [–0.05; 0.02] |
| | Distance 2: Session 2: Condition 1 | 0.01 | 0.02 | 0.37 | 1136 | 0.71 | [–0.03; 0.04] |
| | Distance 3: Session 2: Condition 1 | –0.01 | 0.02 | –0.31 | 1136 | 0.75 | [–0.04; 0.03] |
| Distance x Session x Condition | (against *Distance* 4, *Session* 3 and *Condition* 2) | | | | | | |

