## [Editor Report · eLife Assessment]

This study provides a **valuable** examination of the social discrimination abilities of a jumping spider, Phippidus regius, based on visual cues. Behavioral essays yielded **solid** evidence that these spiders discriminate between familiar and unfamiliar individuals on the basis of visual cues, however the experimental support for individual recognition and long-term memory is **incomplete**. While the results supply evidence of discrimination, additional experiments would be needed to verify the evidence of individual recognition.

---

## [Referee Report · Reviewer #1 (Public review)]

Summary:

The paper sets out to examine the social recognition abilities of a 'solitary' jumping spider species. It demonstrates that based on vision alone spiders can habituate and dishabituate to the presence of conspecifics. The data support the interpretation that these spiders can distinguish between conspecifics on the basis of their appearance.

Strengths:

The study presents two experiments. The second set of data recapitulates the findings of the first experiment with a independent set of spiders, highlighting the strength of the results. The study also uses a highly quantitative approach to measuring relative interest between pairs of spiders based on their distance.

Weaknesses:

The study design is overly complicated, while missing key controls, and the data presented in the figures are not clearly connected to study. The discussion is challenging to understand and appears to make unsupported conclusions.

(1) Study design: The study design is rather complicated and as a result it is difficult to interpret the results. The spiders are presented with the same individual twice in a row, called a habituation trial. Then a new individual is presented twice in a row. The first of these is a dishabituation trial and the second another habituation trial (but now habituating to a second individual). This done with three pairings and then this entire structure is repeated over three sessions. The data appear to show the strong effects of differences between habituation and dishabituation trials in the first session. The decrease in differential behavior between the so-called habituation and dishabituation trials in sessions 2 and 3 are explained as a consequence of the spiders beginning to habituate in general to all of the individuals. The claim that the spiders remember specific individuals is somewhat undercut because all of the 'dishabituation' trials in session 2 are toward spiders they already met for 14 minute previously but seemingly do not remember in session 2. In session 3 it is ambiguous what is happening because the spiders no longer differentiate between the trial types. This could be due to fatigue or familiarity. A second experiment is done to show that introducing a totally novel individual, recovers a large dishabituation response, suggesting that the lack of differences between 'habituation' and 'dishabituation' trials in session 3 is the result of general habituation to all of the spiders in the session rather than fatigue. As mentioned before, these data do support the claim that the spiders differentiate among individuals.

The data from session 1 are easy to interpret. The data from sessions 2 and 3 are harder to understand, but these are the trials in which they meet an individual again after a substantial period of separation. Other studies looking at recognition in ants and wasps (cited by the authors) have done a 4 trial design in which focal animal A meets B in the first trial, then meet C in the second trial, meets B again in the third trial, and then meets D in the last trial. In that scenario trials 1, 2 and 4 are between unfamiliar individuals and trial 3 is between potentially familiar individuals. In both the ants and wasps, high aggression is seen in species with and without recognition on trial 1, with low aggression specifically for trials with familiar individuals in species with recognition. Across different tests, species or populations that lack recognition have shown a general reduction in aggression towards all individuals that becomes progressively less aggressive over time (reminiscent of the session 2 and 3 data) while others have maintained modest levels of aggression across all individuals. The 4 session design used in those other studies provides an unambiguous interpretation of the data, while controlling for 'fatigue'. That all trials in sessions 2 and 3 are always with familiar individuals make it challenging to understand how much the spiders are habituating to each other versus having some kind of associative learning of individual identity and behavior.

The data presentation is also very complicated. How is it the case that a negative proportion of time is spent? The methods reveal that this metric is derived by comparing the time individuals spent in each region relative to the previous time they saw that individual. At the very least, data showing the distribution of distances from the wall would be much easier to interpret for the reader.

(2) "Long-term social memory": It is not entirely clear what is meant by the authors when they say 'long-term social memory', though typically long-term memory refers to a form of a memory that require protein synthesis. While the precise timing of memory formation varies across species and contexts, a general rule is that long term memory should last for > 24 hours (e.g., Dreier et al 2007 Biol Letters). The longest time that spider are apart in this trial set up is something like an hour. There is no basis to claim that spiders have long term social memory as they are never asked to remember anyone after a long time apart. The odd phrasing of the 'long-term dishabutation' trial makes it seem that it is testing a long-term memory, but it is not. The spiders have never met. The fact that they are very habituated to one set of stimuli and then respond to a new stimulus is not evidence of long-term memory. To clearly test memory (which is the part really lacking from the design), the authors would need to show that spiders - upon the first instance of re-encountering a previously encountered individual are already 'habituated' to them but not to some other individuals. The current data suggest this may be the case, but it is just very hard to interpret given the design does not directly test memory of individuals in a clear and unambiguous manner.

(3) Lack of a functional explanation and the emphasis on 'asociality': It is entirely plausible that recognition is pleitropic byproduct of the overall visual cognition abilities in the spiders. However, the discussion that discounts territoriality as a potential explanation is not well laid out. First, many species that are 'asocial' nevertheless defend territories. It is perhaps best to say such species are not group living, but they have social lives because they encounter conspecifics and need to interact with them. Indeed, there are many examples of solitary living species that show the dear enemy effect, a form of individual recognition, towards familiar territorial neighbors. The authors in this case note that territorial competition is mediated by the size of color of the chelicerae (seemingly a trait that could be used to distinguish among individuals). Apparently because previous work has suggested that territorial disputes can be mediated by a trait in the absence of familiarity has led them to discount the possibility that keeping track of the local neighbors in a potentially cannibalistic species could be a sufficient functional reason. In any event, the current evidence presented certainly does not warrant discounting that hypothesis.

Comments on Revision:

The authors have not actually addressed my points and their comments conflate discrimination with recognition. The extensive discussion about how babies are tested for discrimination tasks in their rebuttal misses the point. I believe that the data do show that the spiders discriminate between individuals but whether individuals are recognized (i.e., remembered) is less clear. The authors defend their convoluted study design, but it is overly complex and challenging to interpret the data as a result.

The main issue with the design is that they do not actually test for any kind of memory of specific individuals after a substantial time of separation. Instead they show that a new individuals is still surprising/dishabituating. That is nice evidence for discrimination but does not show memory in a clear and unambiguous way.

My comments and critique are unchanged since they didn't really change the paper. New experiments were needed and they didn't do any. Perhaps it is hard to get the spiders where they are? I don't really understand why they didn't do additional experiments as part of this revision.

---

## [Referee Report · Reviewer #3 (Public review)]

Summary:

Jumping spiders (family Salticidae) have extraordinarily good eyesight, but little is known about how sensitive these small animals might be to the identity of other individuals that they see. Here, experiments were carried out using Phidippus regius, a salticid spider from North America. There were three steps in the experiments; first, a spider could see another spider; then its view of the other spider was blocked; and then either the same or a different individual spider came into view. Whether it was the same or a different individual that came into view in the third step had a significant effect on how close together or far apart the spiders positioned themselves. It has been demonstrated before that salticids can discriminate between familiar and unfamiliar individuals while relying on chemical cues, but this new research on P. regius provides the first experimental evidence that a spider can discriminate by sight between familiar and unfamiliar individuals.

Clark RJ, Jackson RR (1995) Araneophagic jumping spiders discriminate between the draglines of familiar and unfamiliar conspecifics. Ethology, Ecology and Evolution 7:185-190

Strengths:

This work is a useful step toward a fuller understanding of the perceptual and cognitive capacities of spiders and other animals with small nervous systems. By providing experimental evidence for a conclusion that a spider can, by sight, discriminate between familiar and unfamiliar individuals, this research will be an important milestone. We can anticipate a substantial influence on future research.

Weaknesses:

(1) The conclusions should be stated more carefully.

(2) It is not clearly the case that the experimental methods are based on 'habituation (learning to ignore; learning not to respond). Saying 'habituation' seems to imply that certain distances are instances of responding and other distances are instances of not responding but, as a reasonable alternative, we might call distance in all instances a response. However, whether all distances are responses or not is a distracting issue because being based on habituation is not a necessity.

(3) Besides data related to distances, other data might have been useful. For example, salticids are especially well known for the way they communicate using distinctive visual displays and, unlike distance, displaying is a discrete, unambiguous response.

(4) Methods more aligned with salticids having extraordinarily good eyesight would have useful. For example, with salticids, standardising and manipulating stimuli in experiments can be achieved by using mounts, video playback and computer-generated animation.

(5) An asocial-versus-social distinction is too imprecise, and it may have been emphasised too much. With P. regius, irrespective of whether we use the label asocial or social, the important question pertains to the frequency of encounters between the same individuals and the consequences of these encounters.

(6) Hypotheses related to not-so-strictly adaptive factors are discussed and these hypotheses are interesting, but these considerations are not necessarily incompatible with more strictly adaptive influences being relevant as well.

Comments on Revision:

The authors have responded reasonably to the comments I made. There is nothing else that I wish to add.

---

## [Author Response]

The following is the authors’ response to the original reviews.

**Reviewer #1 (Public Review):**
Summary:The paper sets out to examine the social recognition abilities of a 'solitary' jumping spider species. It demonstrates that based on vision alone spiders can habituate and dishabituate to the presence of conspecifics. The data support the interpretation that these spiders can distinguish between conspecifics on the basis of their appearance.

We appreciate the reviewer’s summary. We indeed aimed at investigating the social recognition abilities of the solitary jumping spider (Phidippus regius), using visual cues alone. By employing a habituation-dishabituation paradigm, well-established in developmental psychology, we found support for the interpretation that these spiders can distinguish between conspecifics based on their appearance, as the reviewer noted.

Strengths:The study presents two experiments. The second set of data recapitulates the findings of the first experiment with an independent set of spiders, highlighting the strength of the results. The study also uses a highly quantitative approach to measuring relative interest between pairs of spiders based on their distance.

We appreciate the reviewer's acknowledgement of the strengths of our study. The second set of data underscores the robustness and reliability of the results. Additionally, however, the second experiment served the purpose of disentangling whether the habituation effect observed over sessions was caused by ‘physical’ or ‘cognitive’ fatigue by employing ‘long-term’ dishabituation trials at the end of Session 3. These trials are critical in our study as they help to differentiate between recognition of individual identities versus recognition of familiar individuals (as opposed to unfamiliar ones) and to determine if the observed effects are due to ‘general habituation’ or ‘specific recognition’. We will elaborate on this further below in this revision.

As stated by the reviewer, we employed a highly quantitative approach to measure relative interest between pairs of spiders based on their distance, providing precise and objective data to support our conclusions.

Weaknesses:The study design is overly complicated, missing key controls, and the data presented in the figures are not clearly connected to the study. The discussion is challenging to understand and appears to make unsupported conclusions.

While we acknowledge that the study design is indeed complex, this complexity is essential for conducting a well-controlled and balanced experiment regarding the experimental conditions.

The habituation-dishabituation paradigm is a well-established paradigm in developmental psychology with non-verbal infants. It is understood that during the habituation phase, an individual's attention to a repeated stimulus decreases as they engage in information processing and form a mental representation of it. As the stimulus becomes familiar, it loses its novelty and interest. When a new stimulus is introduced, a recovery of attention suggests that the individual has compared this new stimulus to the stored memory of the habituation stimulus and detected a difference. This process suggests that the individual not only remembered the original stimulus but also recognized the new one as distinct (for a review Kavšek & Bornstein, 2010).

This paradigm has also been extensively applied in animal research, where, like infants, nonverbal subjects rely on recognition and discrimination processes to demonstrate their cognitive abilities. The use of this paradigm dates back to seminal studies such as Humphrey (1974), which explored the perceptual world of monkeys, illustrating how species and individuals are perceived and recognized. In another previous study (Dahl, Logothetis, and Hoffman, 2007), we utilized an even more complex experimental design that incorporated dedicated baseline trials for both habituation and dishabituation phases, which was well-received despite its complexity. In the current study, we contrast dishabituation and habituation trials directly, creating a sequential cascade where each trial is evaluated against the preceding one as its baseline.

On the basis of these arguments, we respectfully decline the claim that this paradigm is inappropriate or lacks key controls. Our study design, though complex, is rigorously grounded in established methodologies and offers a robust framework for exploring individual recognition in Phidippus regius.

However, we take the reviewer’s comments seriously and are committed to identifying and addressing the aspects in our manuscript that may have led to misunderstandings. We clarify these areas in our revision of the manuscript. Modifications were made in the Introduction, Methods, and Discussion sections.

Dahl, C. D., Logothetis, N. K., & Hoffman, K. L. (2007). Individuation and holistic processing of faces in rhesus monkeys. Proceedings of the Royal Society B: Biological Sciences, 274(1622), 2069-2076.

Humphrey, N. K. (1974). Species and individuals in the perceptual world of monkeys. Perception, 3(1), 105-114.

Kavšek, M., & Bornstein, M. H. (2010). Visual habituation and dishabituation in preterm infants: A review and meta-analysis. Research in developmental disabilities, 31(5), 951-975.

(1) Study design: The study design is rather complicated and as a result, it is difficult to interpret the results. The spiders are presented with the same individual twice in a row, called a habituation trial. Then a new individual is presented twice in a row. The first of these is a dishabituation trial and the second is another habituation trial (but now habituating to a second individual). This is done with three pairings and then this entire structure is repeated over three sessions.

While we acknowledge that the design is complex, this complexity is essential for conducting a well-controlled experiment, as described earlier. As the reviewer noted, our design involves presenting the same individual to the focal spider twice in a row (habituation trial), followed by a new individual (dishabituation trial), and then repeating this structure. This approach is fundamental to the habituation-dishabituation paradigm, which allows us to systematically compare the responses to a familiar individual with those elicited by a novel one. If the spiders exhibit different behaviours in terms of the distance they maintain when encountering the same individual versus a new one, it indicates that they are processing the stimuli differently, consistent with recognition memory. This differential response is a key indicator that the spiders can distinguish between familiar and unfamiliar individuals, demonstrating not only a decrease in interest or engagement due to repeated exposure but also a cognitive process where the lack of a matching memory template triggers a distinct behavioural response when confronted with novel stimuli.

By repeating this sequence two more times (Session 2 and 3), we aim to assess the consistency of this recognition process over time. If the focal spider does not remember the individuals from the previous session (one hour ago), we expect consistent behavioural responses across sessions. Conversely, if there is a decrease in response magnitude but the overall response patterns are maintained, we can infer that the focal spider recognizes the previously presented individuals and exhibits habituation, reflected in reduced response intensity. In other words, over sessions and repeated exposure to the same individuals, the memory traces become more firmly established, leading to a situation where a dishabituation trial introduces less novelty, as the spider's recognition of previously encountered individuals becomes more robust and consistent to the point where “habituation” and “dishabituation” trials become indistinguishable, as observed in Session 3. This method allows us to assess the duration of identity recognition in these spiders, indicating how long the memory of specific individuals persists.

All of these outcomes were anticipated before we began Experiment 1. Given that the results aligned with our predictions, we then sought to determine whether the observed reduction in the magnitude of the effect (i.e., the difference between habituation and dishabituation trials) was due to a physical fatigue effect, where the spiders might simply be getting tired, or a cognitive fatigue effect, where the spiders recognized the individuals and as a result did not exhibit any novelty response. To address this, we replicated the experiment with a new group of spiders and introduced special (long-term dishabituation) trials at the end, where the focal spider was presented with a novel spider.

These extra trials allowed us to disentangle the nature of the diminishing response across repeated sessions: a lack of dishabituation (remaining distant) would suggest general physical fatigue, whereas a strong dishabituation response (approaching closely) to the novel spider would indicate cognitive fatigue, thereby confirming that the spiders were indeed recognizing the familiar individuals throughout the experiment.

In light of these considerations, we believe that the complexity of our design is not only justified but absolutely necessary to rigorously test the cognitive capabilities of the spiders. Nonetheless, we understand the need for clarity in presenting our findings and are committed to refining our manuscript to better communicate the rationale and results of our study.

The data appear to show the strong effects of differences between habituation and dishabituation trials in the first session. The decrease in differential behavior between the socalled habituation and dishabituation trials in sessions 2 and 3 is explained as a consequence of the spiders beginning to habituate in general to all of the individuals.

The key question, as mentioned above, is to determine the underlying cause of this general habituation across sessions. Specifically, we aim to differentiate between two potential causes: physical fatigue, where the spiders may simply become less responsive due to the demands of the three-hour testing period, or cognitive fatigue, where the repeated exposure to the same individuals leads to a decreased response because the spiders have started to recognize these individuals over multiple repetitions.

To address this, we replicated the experiment and introduced each focal spider to a new individual in what we termed "long-term dishabituation" trials. By comparing the spiders' responses to these novel individuals with their responses in earlier trials, we sought to better understand the underlying mechanisms of habituation and the duration of individual recognition. The strong dishabituation response observed in these trials is indicative of cognitive fatigue, supporting the presence of recognition memory rather than a general physical fatigue effect.

The claim that the spiders remember specific individuals is somewhat undercut because all of the 'dishabituation' trials in session 2 are toward spiders they already met for 14 minutes previously but seemingly do not remember in session 2.

We appreciate the reviewer’s comment regarding the claim that spiders do not remember specific individuals. This assessment does not align with the rationale of our experiment. The reviewer noted that the dishabituation trials in session 2 involved spiders previously encountered and suggested that the lack of a clear memory response might undercut the claim of specific individual recognition.

However, as we explained earlier, we expect habituation in Session 2 relative to Session 1 precisely because spiders recognize each other in Session 2. If there were no such habituation in Sessions 2 or 3, it would suggest that the spiders’ recognition memory does not persist beyond one hour.

Additionally, it is important to correct the timing noted by the reviewer: each individual spider reencounters the same spider exactly one hour later, not 14 minutes. This is detailed in Table 2 of the manuscript, which outlines that each trial lasts 7 minutes, with a 3-minute visual separation between trials. With six trials per session, this totals to 1 hour per session. Thus, every pair of spiders re-encounters exactly 1 hour after their last interaction.

Again, it is important to clarify that the observed decrease in differential behaviour is not indicative of a failure to remember specific individuals. Rather, it reflects a systematic pattern of habituation, which is a common and expected outcome in such paradigms. This systematic decrease in response strength suggests that the spiders recognize the previously encountered individuals and becoming less responsive over repeated exposures, consistent with the process of habituation. In different terms, the repeated exposure to the same individuals leads to more firmly established memory traces, leading to a situation where a dishabituation trial introduces less novelty, as the spider's recognition of previously encountered individuals becomes more robust and consistent.

Based on the explanations provided above, we respectfully reject the claim that “the spiders remember specific individuals is somewhat undercut […]”. In contrast, this claim is incorrect, as the exact opposite is true. The very strength of our study lies in demonstrating that spiders possess robust recognition memory, as evidenced by a clear dissociation of habituation and dishabituation trials in Session 1, followed by a gradually diminishing effect over Session 2 and 3 as the spiders are increased exposed to the same individuals: Furthermore, the strong rebound from habituation observed in long-term dishabituation trials, where the spiders were exposed to novel individuals.

This misunderstanding suggests that we should take additional care in the revised manuscript to clarify our explanations and provide more detail, ensuring that the rationale behind our experimental design and findings are communicated effectively.

In session 3 it is ambiguous what is happening because the spiders no longer differentiate between the trial types. This could be due to fatigue or familiarity.

The reviewer proposes that the absence of differentiation between 'habituation' and 'dishabituation' trials in Session 3 might be attributed to either fatigue or familiarity. We interpret "fatigue" as what we have termed the “physical fatigue effect” and "familiarity" as “cognitive fatigue effect.” In this context, we concur with the reviewer’s observation, and this very line of reasoning prompted us to conduct a further experiment following the outcome of Experiment 1.

A second experiment is done to show that introducing a totally novel individual, recovers a large dishabituation response, suggesting that the lack of differences between 'habituation' and 'dishabituation' trials in session 3 is the result of general habituation to all of the spiders in the session rather than fatigue. As mentioned before, these data do support the claim that spiders differentiate among individuals.

As the reviewer rightly noted, we addressed these possibilities in our second experiment by introducing a completely novel individual to the spiders, which resulted in a strong dishabituation response. This outcome suggests that the lack of differentiation in Session 3 is more likely due to cognitive habituation rather than physical fatigue. The robust response to novel individuals demonstrates that the spiders are capable of distinguishing between familiar and unfamiliar individuals, suggesting that the reduced differentiation is a consequence of habituation from repeated encounters with the same individuals.

We appreciate the reviewer's recognition that these findings support the conclusion that spiders are capable of differentiating between individual conspecifics.

Additionally, it is important to clarify the structure of our sessions. Each of the 6 trials lasts 7 minutes with a 3-minute visual separation, resulting in a total of 1 hour per session. This ensures that each pair of spiders is encountered exactly one hour later, which controls for the timing and allows us to evaluate the spiders' recognition memory over repeated sessions.

In summary, while the data show a decrease in differential behaviour between habituation and dishabituation trials in Session 2 and 3, the results from our second experiment support the interpretation that this is due to ‘cognitive habituation’ (familiarization) rather than ‘physical fatigue’ (general habituation). This habituation effect underscores the spiders' ability to recognize and become familiar with specific individuals over time, reinforcing our conclusion that they can differentiate among individuals.

The data from session 1 are easy to interpret. The data from sessions 2 and 3 are harder to understand, but these are the trials in which they meet an individual again after a substantial period of separation.

The data from Session 1 are straightforward to interpret, showing clear differences between habituation and dishabituation trials. However, the data from Sessions 2 and 3 are more complex, as these sessions involve the spiders re-encounter individuals after a 1-hour period of separation. Importantly, the outcome is not an artefact in our experiment, but the consequence of a deliberate choice in the experimental design to assess whether spiders can recognise each other after this duration. We believe that this complexity aligns with our expectations, based on the assumption that spiders can recognise each other after one hour. The observed pattern of habituation in Sessions 2 and 3 suggests that the spiders retain memory of the individuals, leading to decreased responsiveness upon repeated encounters. This interpretation is further supported by the Experiment 2, which introduced a novel individual and elicited a strong dishabituation response. This finding confirms that the reduced differentiation in later sessions is due to cognitive habituation rather than physical fatigue, supporting the conclusion that recognition memory last at least one hour.

We hope this explanation clarifies our findings and the rationale behind our relatively complex experimental design choice.

Other studies looking at recognition in ants and wasps (cited by the authors) have done a 4 trial design in which focal animal A meets B in the first trial, then meets C in the second trial, meets B again in the third trial, and then meets D in the last trial. In that scenario trials 1, 2, and 4 are between unfamiliar individuals and trial 3 is between potentially familiar individuals. In both the ants and wasps, high aggression is seen in species with and without recognition on trial 1, with low aggression specifically for trials with familiar individuals in species with recognition. Across different tests, species or populations that lack recognition have shown a general reduction in aggression towards all individuals that become progressively less aggressive over time (reminiscent of the session 2 and 3 data) while others have maintained modest levels of aggression across all individuals. The 4 session design used in those other studies provides an unambiguous interpretation of the data while controlling for 'fatigue'.

We acknowledge that there are multiple ways to design experiments to test recognition memory. In fact, we considered using the paradigm similar to the one proposed by the reviewer and used in studies like Dreier et al., which involves a series of trials with unfamiliar and familiar individuals over extended intervals. We then, however, opted for a more complex design to rigorously assess how habituation and recognition memory develop over repeated sessions with shorter intervals.

In the following, we would like to describe the advantages and disadvantages of both paradigms and outline how we ended up using the more complex version:

Advantages of our paradigm:

As pointed out, by repeating the sequence in exactly similar manner (every same pair of spiders reoccurs after exactly 1 and 2 hours), we can comprehensively evaluate the effect of habituation over multiple exposures. This allows us to assess the extent of the spiders’ memory, when a spider shows stronger habituation to individuals that were novel in Session 1 but “familiar” by the time they encounter them again in Session 2. To achieve this, we need to ensure that each trial and visual separation is precisely timed, ensuring consistent intervals between encounters. As a consequence, each individual spider undergoes the exact same experimental protocol. Most critically, however, are the novel individuals presented after Session 3 (long-term dishabituation trials) that help differentiate between cognitive habituation and physical fatigue. Disadvantages of our paradigm:

The sequences of habituation and dishabituation trials may make the design more complex, as pointed out by the reviewer. As a consequence, the interpretation will become more difficult. However, the data perfectly align with our predictions, and the outcomes were as anticipated in two independently run experiments with two groups of spiders. This highlights the reliability of our experimental design and robustness of our findings.

Advantages of the 4-trial paradigm proposed by the reviewer:

Clearly, the structure of the proposed design is simpler, making interpretation easier. The paradigm also accommodates longer intervals between trials (e.g., 24 hours). Longer intervals could theoretically have been applied in our study. (However, we chose not to leave the spiders in the experimental box longer than necessary, opting instead to return them to their home containers for the night to ensure their well-being. And, a 24-hour interval targets a different phase in the process of long-term memory, but more to this topic further below.)

Disadvantages of the 4-trial paradigm proposed by the reviewer:

Strictly replicating the 4-trial design would result in one familiar encounter versus three unfamiliar ones. This imbalance might introduce bias and limit the robustness of the measurements. Additionally, the design provides less data overall, as the focal individual will be confronted with three other individuals, who will then be excluded from further testing as focal subjects themselves. In contrast, our design ensures a balanced number of familiar0020(habituation) and novel encounters (dishabituation) for each focal individual, allowing for more efficient and comprehensive data collection without excluding individuals from further testing.

Given the aforementioned considerations, we determined that the advantages of our experimental design, in particular the assessment of a cognitive fatigue effect when encountering the same individuals again, outweigh those of the proposed 4-trial design. The mentioned limitations of the 4-trial design, such as the potential for bias and less comprehensive data collection, do not justify re-running the study, especially when the best case scenario is fewer insights than our already existing findings. Our current paradigm yielded results that align perfectly with our predictions, offering a thorough and reliable understanding of recognition memory and habituation in spiders. Therefore, we believe our approach provides a more complete and robust answer to our research questions.

However, we acknowledge that there might be insufficient information in the manuscript addressing the rationale behind our design choices, and we will revise the manuscript to provide a clearer explanation of why our approach is well suited to answering the research questions at hand.

That all trials in sessions 2 and 3 are always with familiar individuals makes it challenging to understand how much the spiders are habituating to each other versus having some kind of associative learning of individual identity and behavior.

We understand the reviewer's concern that having all trials in Sessions 2 and 3 involve familiar individuals could make it challenging to distinguish between general habituation and associative learning of individual identities. In our study, we contrast habituation and dishabituation trials: If general habituation were occurring, we would expect uniformly reduced responses (around the zero line) to all individuals over time, indicating that the spiders are getting used to any individual regardless of their specific identity. However, this is not the case. Our data show that while the responses in Session 2 are reduced in effect size compared to Session 1, they are not flat (around the zero line). This indicates that the spiders still differentiate between a repetition of a spider identity (habituation trials) and two different spider identities (dishabituation trials), albeit with a reduced response strength. The systematicity in the data suggests that the spiders are not merely habituating to any individual, but are instead retaining some level of recognition between specific individuals.

Only by Session 3 do the spiders fully habituate to the point where the responses to habituation and dishabituation trials converge, indicating a complete habituation effect. The introduction of novel individuals in our long-term dishabituation trials further supports the idea that the spiders are recognizing specific individuals rather than exhibiting general habituation. If the spiders were experiencing general habituation, we would not expect the strong dishabituation response observed in our study.

The data presentation is also very complicated. How is it the case that a negative proportion of time is spent? The methods reveal that this metric is derived by comparing the time individuals spent in each region relative to the previous time they saw that individual.

We understand the reviewer's concern regarding the complexity of the data presentation and the calculation of the negative proportion of time. Regarding the complexity of the design, we have already justified our choice of a more intricate experimental setup. This complexity is necessary for accurately assessing recognition memory and habituation over repeated sessions.

The metric is derived by comparing the time individuals spent in each region (relative to the transparent front panel) in the current trial (n) relative to the previous trial (n-1). With multiple trials, this results in a cascade of trials and conditions. This method was established in

Humphrey’s and our previous study (Humphrey, 1974; Dahl, Logothetis, Hoffman, 2007), where we demonstrated its effectiveness in assessing individuation of faces in macaque monkeys.

Also in our current experimental design, each current trial is contrasted with the preceding one, allowing us to compare distributions of distances taken in two trials. In this context, every preceding trial serves as baseline for every current trial.

Figure 1 of the manuscript, illustrates the structure and analysis of the trials,

Panel a depicts the baseline, habituation, and dishabituation trials, where spiders are exposed to different conspecifics.

Baseline (left panel, red): When two spiders are visually exposed to each other for the first time, it is expected that they will explore each other closely, exhibiting high levels of proximity (initial exploratory behaviour).

Habituation (centre panel, green): When the same spiders are reintroduced in a subsequent round of exposure, it is anticipated that they will exhibit reduced exploratory behaviour and maintain a greater distance compared to the baseline trial, if they recognize each other from the previous encounter (indicative of habituation).

Panel b (upper and middle panels; red and green): Demonstrates the theoretical assumptions and expected changes in behaviour:

By subtracting the distribution of distances in the baseline trial from the habituation trial, we generate a delta distribution. This delta distribution reveals negative values near the transparent panel (indicating reduced proximity in the habituation trial) and positive values at mid- to fardistances (indicating increased distancing behaviour). This delta distribution is also what is reported in Figure 2.

Dishabituation: In this trial, a new spider (different from the one in the habituation trial) is introduced. The dishabituation trial will be considered in contrast to the habituation trial described above. If the spider recognizes the new individual as different, it is expected to show increased exploratory behaviour and reduced distance, similar to the initial baseline trial.

By subtracting the distribution of distances in the habituation trial from the dishabituation trial, we obtain another delta distribution. This delta distribution should reveal positive values near the transparent panel (indicating increased proximity in the dishabituation trial) and negative values at mid- to far-distances (indicating decreased proximity compared to the habituation trial).

We hope this clarifies the rationale behind our data presentation and the methodological approach we employed. We have revised the figure to enhance its clarity and make it more intuitive for the reader.

Dahl, C. D., Logothetis, N. K., & Hoffman, K. L. (2007). Individuation and holistic processing of faces in rhesus monkeys. Proceedings of the Royal Society B: Biological Sciences, 274(1622), 2069-2076.

Humphrey, N. K. (1974). Species and individuals in the perceptual world of monkeys. Perception, 3(1), 105-114.

At the very least, data showing the distribution of distances from the wall would be much easier to interpret for the reader.

We understand the reviewer's concern that data showing the distribution of distances from the wall would be much easier to interpret for the reader. We initially consider that but came to the conclusion that this approach is not straightforward. For instance, if both spiders are positioned at the very front but in different corners, the distance to the panel would be very small, but the distance between the spiders would be large. Thus, using distances from the wall could misrepresent the actual spatial distribution between the spiders.

(2) "Long-term social memory": It is not entirely clear what is meant by the authors when they say 'long-term social memory', though typically long-term memory refers to a form of a memory that requires protein synthesis.

To address this conceptually, we used the term "long-term social memory" to describe the spiders' ability to recognize and remember individual conspecifics over multiple experimental sessions. While social memory refers to the ability of an individual to recognize other individuals within a social context, long-term memory typically involves the retention of information over extended periods. Recognizing that the term “long-term social memory” is not commonly used, we have revised the manuscript to use the more standard term “long-term memory.”

While the precise timing of memory formation varies across species and contexts, a general rule is that long-term memory should last for > 24 hours (e.g., Dreier et al 2007 Biol Letters). The longest time that spiders are apart in this trial setup is something like an hour. There is no basis to claim that spiders have long-term social memory as they are never asked to remember anyone after a long time apart.

We appreciate the reviewer’s feedback regarding the term "long-term social memory." The statement "long-term memory should last for > 24 hours" is a generalisation in discussions about memory. It oversimplifies a more complex topic. That is, long-term memory is typically distinguished from short-term memory by its persistence over time, often lasting from hours to a lifetime. However, the exact duration that qualifies memory as "long-term" varies depending on the context, model species, and type of memory. In studies involved in synaptic plasticity (LTP), the object might indeed be to look at memory that persists for at least 24 hours as a criterion for long-term memory. In studies of cellular and/or molecular mechanisms where the stabilization and consolidation of memory traces over time are key areas of interest this 24-hour interval is very common. But, defining long-term memory strictly by a 24-hour duration is by no means universally accepted nor does it apply across all fields of study.

To clarify, long-term memory is a process involving consolidation starting within minutes to hours after learning. Clearly, full consolidation can take longer, while memory persisting 24 hours is considered fully consolidated. But this does not mean that memory lasting less than 24 hours are not part of long-term memory.

In fact, Atkinson and Shiffrin (1969) proposed that information entering short-term memory remains there for about 20 to 30 seconds before being displaced due to space limitations. During this brief interval, initial encoding processes begin transferring information to long-term memory, establishing an initial memory trace. This transfer is not indicative of full consolidation but represents the initial "laying down" of the memory trace (encoding). In our study, the focal spider’s brain forms initial memory traces of the individuals it encounters. This process continues during the period of visual separation. Upon re-encountering the same individual a few minutes later, the spider accesses the initial memory trace stored in long-term memory. This trace is fragile and not fully consolidated. The re-encounter acts as a rehearsal, reactivating specific memory traces and potentially strengthening them through additional encoding processes, allowing the spider to recognize the individual even an hour later.

According to Markowitsch (2013), initial encoding in long-term memory begins within seconds to minutes. It is also important to note that we argue for identity recognition rather than identity recall. Recognition involves correctly identifying a stimulus when it is presented again, while recall requires the volitional generation of information without an external stimulus. Thus, recall may rely on deeper forms of memory consolidation than recognition.

Is protein synthesis required for long-term memory?

The role of protein synthesis in long-term memory has been extensively studied. According to Castellucci et al. (1978), explicit memory comprises a short-term phase that does not require protein synthesis and a long-term phase that does. Hebbian learning in its initial phase (early LTP) does not necessarily require protein synthesis. This phase involves the rapid strengthening of synapses through existing proteins and signaling pathways, such as the activation of NMDA receptors and the influx of Ca2+ ions. For the changes to persist (late LTP), protein synthesis is important. This phase involves the production of new proteins that contribute to long-term structural changes at the synapse, such as the growth of new synaptic connections or the stabilization of existing ones.

This differentiation between the early and late phases of LTP highlights that long-term memory can begin forming without immediate protein synthesis. Our study focuses on this early phase of memory encoding, which involves the initial formation of memory traces that do not yet depend on protein synthesis.

It is however worth noting that recent research suggests that there is an early phase of protein synthesis (within minutes to hours) through the activation of immediate early genes (IEGs) and transcription factors. In this context, protein synthesis supports initial synaptic modifications. What the reviewer refers to is the consolidation phase (late phase), where continued synthesis of proteins induces structural changes at synapses, leading to the formation of new synaptic connections. In our study, it is plausible to assume that an early form of protein synthesis may contribute to stabilizing the initial memory traces during the encoding phase. However, whether or not protein synthesis occurred in our spiders is beyond the scope of this investigation and was not specifically addressed.

The critical aspect of our study is that the information transitioned from short-term memory to long-term memory during an early encoding phase, allowing recall after an hour. Due to the inherent limitations and transient nature of the short-term memory, it is implausible for spiders to retain these memory representations solely within the short-term memory for such durations. Our findings suggest that the initial encoding processes were robust enough to transfer these experiences into long-term memory, where they were stabilized and could be accessed later.

In sum, it is important to note that long-term memory is a dynamic process, and while testing after 24 hours is a convention in some studies, this timing is arbitrary and not universally applicable to all contexts or species. The more critical consideration here is that we are dealing with a species where no prior evidence of long-term memory exists. Debating a 24-hour delay or the specifics of protein synthesis, while potentially interesting for future studies, detracts from the true significance of our findings. Our study is the first to show something akin to long-term memory representations in this species and this should remain in our focus.

Shiffrin, R. M., & Atkinson, R. C. (1969). Storage and retrieval processes in long-term memory. Psychological review, 76(2), 179.

Markowitsch, H. J. (2013). Memory and self–Neuroscientific landscapes. International Scholarly Research Notices, 2013(1), 176027.

Castellucci, V. F., Carew, T. J., & Kandel, E. R., 1978. Cellular analysis of long-term habituation of the gill-withdrawal reflex of Aplysia californica. Science, 202(4374), 1306-1308.

The odd phrasing of the 'long-term dishabutation' trial makes it seem that it is testing a longterm memory, but it is not. The spiders have never met. The fact that they are very habituated to one set of stimuli and then respond to a new stimulus is not evidence of long-term memory. To clearly test memory (which is the part really lacking from the design), the authors would need to show that spiders - upon the first instance of re-encountering a previously encountered individual are already 'habituated' to them but not to some other individuals. The current data suggest this may be the case, but it is just very hard to interpret given the design does not directly test the memory of individuals in a clear and unambiguous manner.

While we appreciate the reviewer's feedback, we believe there may have been some misunderstanding regarding the term “long-term dishabituation.” The introduction of novel individuals at the end of Session 3 was not intended to test long-term memory by having spiders recognize these novel individuals. Instead, it aimed to investigate the nature of the habituation observed over the three sessions.

The novel individuals introduced at the end of Session 3 serve the purpose to differentiate between general habituation (a decline in response due to repeated exposure to any stimuli) and specific habituation (recognition and reduced response to previously encountered individuals). The novel spiders have never been encountered before, so the focal spiders cannot have prior representations of them. Thus, the strong dishabituation response to these novel individuals indicates that the habituation observed earlier is not due to a general fatigue effect or loss of interest but rather a specific habituation effect to the familiar individuals. By showing such strong and increased response to novel individuals, the study demonstrates that the spiders' increasingly reduced responses in Sessions 2 and 3 are not merely due to a general decrease in responsiveness but suggest cognitive habituation. This cognitive habituation implies that the spiders remember the familiar individuals (as each of them occurred three times across the three sessions), a process that relies on long-term memory. Therefore, while the novel spiders themselves are not a direct test of long-term memory, the use of these novel spiders helps us infer that the habituation observed over the three sessions is indeed due to the formation of long-term memory traces.

In other words, the organism detects and processes the novel stimulus as different from the habituated one. In our study, if a spider showed a strong dishabituation response to a novel individual introduced at the end of Session 3, it would indicate that the spider had formed specific representations of the individuals they encountered during the three sessions. These representations allow the spiders to recognise the novel individuals as different, leading to renewed interest and a stronger behavioural response. It is the absence of a prior representation for the novel spiders that triggers this dishabituation response. Since the novel spider does not match any stored representations of the previously encountered spiders, the focal spider responds more strongly.

The introduction of novel individuals at the end of Session 3 helps clarify that the increasing habituation observed in Session 2 and 3 is specific to familiar individuals, indicating cognitive habituation. This supports the presence of long-term memory processes in the spiders, as they can distinguish between previously encountered individuals and new ones. The habituationdishabituation paradigm thus effectively demonstrates the spiders' ability to form and reactivate encoded memory traces, providing clear evidence of recognition memory.

For these reasons, we are convinced that our interpretation is accurate and hope this clarification renders the additional request for an entirely new experiment unnecessary.

(3) Lack of a functional explanation and the emphasis on 'asociality': It is entirely plausible that recognition is a pleitropic byproduct of the overall visual cognition abilities in the spiders.

We agree with the reviewer that it is essential to consider the broader context of individual recognition and its potential adaptive significance. The possibility that recognition in jumping spiders could be a pleiotropic byproduct of their advanced visual cognition abilities is indeed a plausible explanation and has been discussed in our manuscript.

However, the discussion that discounts territoriality as a potential explanation is not well laid out. First, many species that are 'asocial' nevertheless defend territories. It is perhaps best to say such species are not group living, but they have social lives because they encounter conspecifics and need to interact with them.

The reviewer also correctly points out that many 'asocial' species still defend territories and have social interactions. Our use of the term 'asocial' was meant to indicate that jumping spiders do not live in cohesive social groups, but we acknowledge that they do have social lives in terms of interactions with conspecifics. It is more accurate to describe these spiders as non-groupliving, yet socially interactive species. A better term is “non-social” to refer to the jumping spider as a species that do not live in stable social groups and do not exhibit associated behaviours, such as cooperative behaviours. This also would imply that individuals still interact with conspecifics, especially in contexts like mating, territorial disputes or aggression. We, thus, change the term from “asocial” to “non-social” in the manuscript.

Indeed, there are many examples of solitary living species that show the dear enemy effect, a form of individual recognition, towards familiar territorial neighbors. The authors in this case note that territorial competition is mediated by the size or color of the chelicerae (seemingly a trait that could be used to distinguish among individuals). Apparently, because previous work has suggested that territorial disputes can be mediated by a trait in the absence of familiarity has led them to discount the possibility that keeping track of the local neighbors in a potentially cannibalistic species could be a sufficient functional reason. In any event, the current evidence presented certainly does not warrant discounting that hypothesis.

The “dear enemy effect”, where solitary living species recognize and show reduced aggression towards familiar territorial neighbors, is a relevant consideration. This effect demonstrates that individual recognition can have significant functional implications even in species that are not group-living. We will elaborate on this effect in the revised manuscript to provide a more comprehensive discussion.

The reviewer mentioned that territorial disputes can be mediated by the size or color of the chelicerae, potentially serving as a feature for individual recognition. Our intention was not to discount the role of such traits but to highlight that the level of identity recognition we observed represents subordinate classification. This is different from the basic-level classification, such as distinguishing between male and female based on chelicerae colour. While we acknowledge that colour can be an important feature for identity discrimination, our findings suggest that individual recognition in jumping spiders goes beyond simple colour differentiation.

**Reviewer #2 (Public Review):**
Summary:In this manuscript, the authors investigated whether a salticid spider, Phidippus regius, recognizes other individuals of the same species. The authors placed each spider inside a container from which it could see another spider for 7 minutes, before having its view of the other spider occluded by an opaque barrier for 3 minutes. The spider was then either presented with the same individual again (habituation trial) or a different individual (dishabituation trial). The authors recorded the distance between the two spiders during each trial. In habituation trials, the spiders were predicted to spend more time further away from each other and, in dishabituation trials, the spiders were predicted to spend more time closer to each other. The results followed these predictions, and the authors then considered whether the spiders in habituation trials were generally fatigued instead of being habituated to the appearance of the other spider, which may have explained why they spent less time near the other individual. The authors presented the spiders with a different (novel) individual after a longer period of time (which they considered to be a long-term dishabituation trial), and found that the spiders switched to spending more time closer to the other individual again during this trial. This suggested that the spiders had recognized and had habituated to the individual that they had seen before and that they became dishabituated when they encountered a different individual.

We appreciate the reviewer's detailed summary of our study. The reviewer's summary accurately captures the essence of our experimental design, predictions, and findings.

Strengths:It is interesting to consider individual recognition by Phidippus regius. Other work on individual recognition by an invertebrate has been, for instance, known for a species of social wasp, but Phidippus regius is a different animal. Importantly and more specifically, P. regius is a salticid spider, and these spiders are known to have exceptional eyesight for animals of their size, potentially making them especially suitable for studies on individual recognition. In the current study, the results from experiments were consistent with the authors' predictions, suggesting that the spiders were recognizing each other by being habituated to individuals they had encountered before and by being dishabituated to individuals they had not encountered before. This is a good start in considering individual recognition by this species.

We appreciate the reviewer's positive summary and acknowledgment of the strengths of our study. We would like to point out some more details:

While the exceptional eyesight of salticid spiders is indeed a significant factor, our study reaches deeper in terms of processing. We do not argue at the level of sensation rather than at the level of perception. Even more, identity recognition is a higher-level perceptual process. This distinction is crucial: we are not merely examining the spiders' sensory capabilities (such as good eye sight), but rather how their brains interpret and represent what they “see”. This involves a cognitive process where the sensory input (sensation) is processed and integrated into meaningful constructs (perception) and memorised in form of representations.

Our study also suggests that P. regius engages in “higher-level” perceptual processes. This most-likely involves complex representations of individual conspecifics, which in mammalian brains are associated with regions such as the central inferior temporal (cIT) and anterior inferior temporal (aIT) areas. We provide evidence that these spiders do not just sense visual stimuli but interpret and recognize individual identities, indicating sophisticated perceptual and cognitive abilities. In other words, the spiders do not merely respond to visual stimuli in a reflexive manner, but rather engage in sophisticated perceptual and cognitive processes that allow them to recognize and distinguish between individual identities. This indicates that the spiders are not simple Braitenberg vehicles reacting to stimuli, but are thinking organisms capable of complex mental representations. This resonates with current trends in animal cognition research, which increasingly recognize some level of consciousness and advanced cognitive abilities across a wide range of animal species. Moreover, this aligns with the growing interest and recognition of spider cognition, where research begins to provide evidence for the cognitive complexity and perceptual capabilities of these often underestimated creatures (Jackson and Cross, 2011).

Jackson, R. R., & Cross, F. R. (2011). Spider cognition. Advances in insect physiology, 41, 115174.

Weaknesses:The experiments in this manuscript (habituation/dishabituation trials) are a good start for considering whether individuals of a salticid species recognize each other. I am left wondering, however, what features the spiders were specifically paying attention to when recognizing each other. The authors cited Sheehan and Tibbetts (2010) who stated that "Individual recognition requires individuals to uniquely identify their social partners based on phenotypic variation." Also, recognition was considered in a paper on another salticid by Tedore and Johnsen (2013).Tedore, C., & Johnsen, S. (2013). Pheromones exert top-down effects on visual recognition in the jumping spider Lyssomanes viridis. The Journal of Experimental Biology, 216, 1744-1756. doi: 10.1242/jeb.071118In this elegant study, the authors presented spiders with manipulated images to find out what features matter to these spiders when recognizing individuals.

The reviewer raises an important point regarding the specific features that Phidippus regius might be paying attention to when recognizing individual conspecifics. Our study indeed cited Sheehan and Tibbetts (2010) to highlight the importance of phenotypic variation in individual recognition. Additionally, we referenced the work by Tedore and Johnsen (2013) on visual recognition in another salticid species, which suggests that multiple sensory modalities, including visual and pheromonal cues, may be involved in the recognition process. While our current study focused on demonstrating that Phidippus regius can recognize individual conspecifics, we acknowledge that it does not specifically identify the phenotypic features involved in this recognition.

Part of the problem with using two living individuals in experiments is that the behavior of one individual can influence the behavior of the other, and this can bias the results.

We appreciate the reviewer's observation regarding the potential bias introduced by using two living individuals in experiments, as the behaviour of one individual can indeed influence the behaviour of the other. We shared this concern initially; however, the consistency of the data with our hypotheses suggests that this potential bias did not adversely affect the validity of our findings, rendering the concern largely illusory at least in the context of our study.

We opted for the living-individual paradigm for the following reasons:

There is a growing trend in ethological as well as animal cognition research towards more ecologically valid and biologically relevant settings, while simultaneously advancing the precision and quantification of the data collected. This is referred to as computational ethology.

This approach advocates for assessing behaviour in environments that more closely resemble natural conditions, rather than relying solely on sterile and artificial experimental setups. The rationale is that such naturalistic arenas allow animals to exhibit a broader range of behaviours and interactions, providing a more accurate reflection of their cognitive and social abilities. The challenge, however, lies in navigating the inherent tradeoff between the strict control offered by standardized procedures and the ecological validity of more naturalistic interactions.

By allowing two spiders to confront each other, we aimed to capture authentic behavioural responses while maintaining a degree of experimental standardization through the use of a controlled setup. Our approach ensures that the behaviours observed are not merely artifacts of an artificial environment but are representative of genuine social interactions. Also, to minimize potential biases arising from mutual behavioural influences, we employed a controlled and repeatable experimental environment.

We believe that the chosen approach provides a meaningful balance (in the above-mentioned trade-off) between ecological validity and experimental rigour. By combining a standardized environment with the naturalistic interaction of real spiders, we ensured that our findings are both scientifically robust and biologically relevant.

However, this issue can be readily avoided because salticids are well known, for example, to be highly responsive to lures (e.g. dead prey glued in lifelike posture onto cork disks) and to computer animation.

While it is true that salticid spiders are responsive to lures and computer animations, we carefully considered the most appropriate and ecologically valid approach for our study. Our aim was to capture genuine behavioural patterns in a context that closely mimics the natural encounters these spiders experience.

Additionally, creating comparable video stimuli of spiders presents its own set of challenges: Video recordings or computer animations may not fully capture the nuanced behaviours and subtle variations that occur during real-life interactions. There is also a risk that such stimuli could be perceived differently by the spiders, potentially introducing new biases or confounding factors.

Scientific progress is not made by merely relying on previously established paradigms, especially when they may not be suitable for the specific context of a study. While alternative methods like lures or computer animations can be valuable in certain situations, our approach was deliberately chosen to best capture the naturalistic and interactive aspects of spider behaviour.

These methods have already been successful and helpful for standardizing the different stimuli presented during many different experiments for many different salticid spiders, and they would be helpful for better understanding how Phidippus regius might recognize another individual on the basis of phenotypic variation. There are all sorts of ways in which a salticid might recognize another individual. Differences in face or body structure, or body size, or all of these, might have an important role in recognition, but we won't know what these are using the current methods alone. Also, I didn't see any details about whether body size was standardized in the current manuscript.

As mentioned previously, the goal of our study was to demonstrate that identity recognition occurs in spiders. This alone is of significant importance, as it challenges existing assumptions about the cognitive capabilities of small-brained animals. We did not aim at providing a proximate explanation (mechanism) for identity recognition in spiders.

The problem with what the reviewer suggested is this: As long as we do not have conclusive evidence that spiders recognize individual conspecifics, any attempt to design and manipulate stimuli would lack a solid foundation. Without understanding whether spiders have this capability, we cannot make informed decisions about which features or characteristics to manipulate in stimuli. In other words, this uncertainty means we lack a starting point for our assumptions, making it nearly impossible to create stimuli that would be useful or relevant in testing identity recognition.

Additionally, it is nearly impossible to artificially generate a stimulus set that encompasses the natural variance in features that spiders use for visual individuation. There is no guarantee that artificial stimuli, such as lures or computer animations, would capture the relevant features that spiders use in natural interactions.

In other words, the question how Phidippus regius recognizes another individual will be subject of further investigation. In this study, we focus on whether or not they individuate others.

For another perspective, my thoughts turn to a paper by Cross et al.Cross, F. R., Jackson, R. R., & Taylor, L. A. (2020). Influence of seeing a red face during the male-male encounters of mosquito-specialist spiders. Learning & Behavior, 48, 104-112. doi: 10.3758/s13420-020-00411-yThese authors found that males of Evarcha culicivora, another salticid species that is known to have a red face, become less responsive to their own mirror images after having their faces painted with black eyeliner than if their faces remained red. In all instances, the spiders only saw their own mirror images and never another spider, and these results cannot be interpreted on the basis of habituation/dishabituation because the spiders were not responding differently when they simply saw their mirror image again. Instead, it was specifically the change to the spider's face which resulted in a change of behavior. The findings from this paper and from Tedore and Johnsen can help give us additional perspectives that the authors might like to consider. On the whole, I would like the authors to further consider the features that P. regius might use to discern and recognize another individual.

We acknowledge that identifying the specific features used by P. regius for identity recognition is a valuable direction for future research. However, we must emphasise that without first establishing whether spiders are capable of individuating each other, it would be premature and challenging to determine the specific features they rely on for this process. A lack of response to certain features could either suggest that those features are not relevant or, more critically, that the spider does not recognize individual identities at all. Thus, our initial focus on demonstrating identity recognition is essential before delving into the specific cues or characteristics involved.

While the call for addressing the proximate causation of identity recognition in jumping spiders is valid, we need to also reiterate the significance of our findings and why they stand on their own merit:

Our study demonstrates for the first time that Phidippus regius can systematically individuate conspecifics, showing habituation within short intervals (10 minutes) and over longer intervals (1 hour). This behaviour is not due to general habituation or physical fatigue but is a result of cognitive habituation, as illustrated by the spiders' response to novel individuals introduced after repeated encounters with familiarized ones.

What are the implications of this? Our findings indicate that these spiders possess long-term memory and form representations that can be reactivated after an hour. While this is most-likely not fully consolidated memory formation (see our reply to Reviewer 1), it represents an encoded long-term memory. This implies that small-brained animals can remember, represent, and potentially build internal mental images, which are crucial for sophisticated cognitive processing.

**Reviewer #3 (Public Review):**
Summary:Jumping spiders (family Salticidae) have extraordinarily good eyesight, but little is known about how sensitive these small animals might be to the identity of other individuals that they see. Here, experiments were carried out using Phidippus regius, a salticid spider from North America. There were three steps in the experiments; first, a spider could see another spider; then its view of the other spider was blocked; and then either the same or a different individual spider came into view. Whether it was the same or a different individual that came into view in the third step had a significant effect on how close together or far apart the spiders positioned themselves. It has been demonstrated before that salticids can discriminate between familiar and unfamiliar individuals while relying on chemical cues, but this new research on P. regius provides the first experimental evidence that a spider can discriminate by sight between familiar and unfamiliar individuals.Clark RJ, Jackson RR (1995) Araneophagic jumping spiders discriminate between the draglines of familiar and unfamiliar conspecifics. Ethology, Ecology and Evolution 7:185-190

We appreciate the reviewer's comprehensive summary and acknowledgment of the significance of our findings.

Strengths:This work is a useful step toward a fuller understanding of the perceptual and cognitive capacities of spiders and other animals with small nervous systems. By providing experimental evidence for a conclusion that a spider can, by sight, discriminate between familiar and unfamiliar individuals, this research will be an important milestone. We can anticipate a substantial influence on future research.

We appreciate the reviewer’s recognition of the strengths and significance of our study. We are pleased that the reviewer considers our research an important milestone. Our findings indeed suggest that even animals with relatively simple nervous systems can perform complex cognitive tasks, which has substantial implications for the broader study of animal cognition.

As pointed out by the reviewer, we also hope that our study will have a substantial influence on future research. By establishing a methodology and providing clear evidence of visual discrimination, we aim to encourage further investigations into the cognitive abilities of jumping spiders and other arthropods. Future research can build on our findings to explore the specific visual cues and mechanisms involved in individual recognition (as Reviewer 2 pointed out), as well as the ecological and evolutionary implications of these abilities.

Weaknesses:(1) The conclusions should be stated more carefully.

We agree that clarity in our conclusions is paramount. We will revise the manuscript to ensure that our conclusions are presented with precision and appropriately reflect the data. Specifically, we will emphasize the evidence supporting our findings of visual individual recognition and clarify the limitations and scope of our conclusions to avoid any potential overstatements.

(2) It is not clearly the case that the experimental methods are based on 'habituation (learning to ignore; learning not to respond). Saying 'habituation' seems to imply that certain distances are instances of responding and other distances are instances of not responding but, as a reasonable alternative, we might call distance in all instances a response. However, whether all distances are responses or not is a distracting issue because being based on habituation is not a necessity.

We appreciate the reviewer's feedback and understand the concern regarding the use of the term 'habituation.' We agree that all distances maintained by the spiders are active responses and reflect their behavioral decisions based on perception and recognition of the other individual. We recognize that all distances are responses and interpret these as the spiders’ “active decisions”, modulated by their recognition of the same or different individuals.

The terms 'habituation' and 'dishabituation' are used to label trial types for ease of discussion and to describe the expected behavioural modulation.

(3) Besides data related to distances, other data might have been useful. For example, salticids are especially well known for the way they communicate using distinctive visual displays and, unlike distance, displaying is a discrete, unambiguous response.

We appreciate the reviewer’s suggestion to incorporate data on visual displays, which are indeed well-known communication methods among salticids. We agree that visual displays are discrete and unambiguous responses that could provide additional insights into the spiders' recognition abilities.

Our primary focus on distance measurements was driven by the need to quantify behaviour in a continuous and scalable manner, that is, how spiders modulate their proximity based on familiarity with other individuals.

We acknowledge the potential value of including visual display measurments; however, in our study, we aimed to establish a foundational understanding of recognition behaviour through proximity measures first. Also, capturing diplays requires a different experimental paradigm, where the displays are clearly visible and analyzable.

(4) Methods more aligned with salticids having extraordinarily good eyesight would be useful. For example, with salticids, standardising and manipulating stimuli in experiments can be achieved by using mounts, video playback, and computer-generated animation.

There is no doubt that salticids have excellent eyesight. However, our study focuses on higherlevel perceptual processes that require complex brain analysis, not just visual acuity. The goal was to investigate whether spiders can individuate and recognize conspecifics, which involves interpreting visual information and forming long-term representations.

Clearly, methods like video playback and computer animations are useful in controlled settings, where the spider is mounted, but they pose challenges for our specific research question. At this stage of research, we lack precise knowledge of which visual features are critical for individual recognition in spiders, making it difficult to design effective artificial stimuli.

Our primary objective was to determine if spiders can individuate others. Before exploring the proximate mechanisms of how they individuate others, it was essential to establish that they have this capability. This foundational question needed to be addressed before delving into more detailed mechanistic studies.

(5) An asocial-versus-social distinction is too imprecise, and it may have been emphasised too much. With P. regius, irrespective of whether we use the label asocial or social, the important question pertains to the frequency of encounters between the same individuals and the consequences of these encounters.

Our intent was to convey that P. regius does not live in cohesive social groups but does engage in individual interactions that can have significant behavioral consequences. We will revise the manuscript to reduce the emphasis on the asocial-versus-social distinction. As discussed above, we also will change the term “asocial” to “non-social” in the manuscript.

(6) Hypotheses related to not-so-strictly adaptive factors are discussed and these hypotheses are interesting, but these considerations are not necessarily incompatible with more strictly adaptive influences being relevant as well.

We appreciate the reviewer's observation regarding the discussion of hypotheses related to notso-strictly adaptive factors. We agree that our considerations of these factors do not preclude the relevance of more strictly adaptive influences.

We will revise the manuscript to explicitly discuss how our findings can be interpreted in the context of adaptive hypotheses. This will provide a more comprehensive understanding of the evolutionary significance of individual recognition in P. regius. Modifications were made in the Discussion section.

In the following, we comment on issues not mentioned in the “public reviews” section.

**Reviewer #1 (Recommendations For The Authors):**
(1) I would suggest conducting experiments that actually test for recognition memory, as this seems to be a claim that the authors make. Following the ant studies by Dreier cited in this manuscript would be sufficient to test for memory. Given the relative simplicity of the measures being taken (location of spiders), this would seem like a very simple addition that would provide a much stronger and more readily interpreted dataset.

As previously explained in our detailed responses (public reviews), we believe that the current design effectively addresses the questions at hand. Our approach, using a habituationdishabituation paradigm, provides robust evidence for recognition memory within the framework of early long-term memory.

Additionally, we have explained why using the distance to the panel as a measure is not appropriate in this context. Specifically, using such a measure can misrepresent the actual interests of the spiders in each other.

While we acknowledge the merits of the ant studies by Dreier, our current design allows for a detailed understanding of the spiders' recognition capabilities over short (10 min) and slightly longer intervals (up to one hour). This is sufficient to demonstrate the presence of recognition memory without the necessity of further experiments. The observed patterns of habituation and dishabituation responses in our study clearly indicate that the spiders can distinguish between familiar and novel individuals, which supports our claims.

Given these points, we respectfully maintain that the current data and experimental design are adequate to support our findings and provide a comprehensive understanding of recognition memory in Phidippus regius.

(2) The writing is rather impenetrable. The results explain the basic finding in terms of statistical variables rather than simply stating the results. A clear and straightforward statement such as 'the spiders showed reduced interest upon habituation trials, indicating xyz' (and then citing the stats) is preferable to the introduction of results as a statistical model. The statistical model is a means of assessing the results. It is not the result. Describe the data.

We tried to improve that in the current version.

(3) Showing more straightforward data such as distance from the joint barrier would make the paper much easier to understand.This paper has been on bioRxiv for some time and my guess is that it has ended up here because it is having trouble in review. Collecting new data that more directly test the question at hand, presenting the data in a more direct manner, and more critically evaluating your own claims will improve the paper.

While it is true that the paper has been on bioRxiv for a while, this submission marks the first instance where it has undergone peer review. Prior to this, the manuscript was submitted to other journals but was not reviewed.

We hope the explanations provided in the “public reviews” section, along with the revised manuscript, sufficiently clarify our study and its conclusions. We believe the current data robustly address the research questions, and as outlined in our detailed responses, we have critically evaluated our claims and presented the data clearly. Given these clarifications, we do not see the necessity for new experiments as the existing data adequately support our findings. We trust that these revisions and explanations will clarify any misunderstandings.

I am totally sold that the spiders are paying attention to identity at some level. The key now is to understand what that actually means in terms of recognition (i.e. memory of individuals) not just habituation.

We appreciate the reviewer’s emphasis on the distinction between habituation and memorybased individual recognition. As detailed in the preceding discussion, we have taken great care to clarify how our paradigm distinguishes simple habituation effects from true memory for individual identity. We trust that the preceding sections make clear how our findings go beyond simple habituation to establish genuine individual recognition.

**Reviewer #2 (Recommendations For The Authors):**
Aside from the comments in the public review, I have some additional comments that the authors may wish to consider.Numerous times in the manuscript, the authors mentioned that recognizing individuals requires recognition memory. This seems rather obvious, and I wonder if the authors could instead be more precise about what they mean by 'recognition memory'?

Recognition memory refers to the cognitive ability to identify a previously encountered stimulus, an individual, or events as familiar. It involves both encoding and retrieval processes, allowing an organism to distinguish between novel and familiar stimuli. This form of memory is a fundamental component of cognitive functioning and is supported by neural mechanisms that, in the mammal brain, involve the hippocampus and other brain regions associated with memory processing.

In our study, we aimed to test whether Phidippus regius recognizes conspecifics, or, in other words, utilizes recognition memory to distinguish between familiar and unfamiliar conspecifics. With the habituation - dishabituation paradigm, we assessed the spiders' ability to recognize previously encountered individuals and demonstrate memory retention over short (10 min) and extended periods (1 hour).

Encoding: In the initial trial, when a spider encounters an individual for the first time (Figure 1A, “Baseline” or “Dishabituation” for every following trial), it encodes the visual information related to that specific individual. This encoding process involves creating a memory trace of the individual's phenotypic characteristics.

Storage: During the visual separation period, this encoded information is stored in the spider's memory system. The memory trace, though initially fragile, starts to stabilize over the separation period. Whether or not this leads to some form of consolidated memory remains unaddressed. This aspect was highlighted by the first reviewer, but our focus is on the early process rather than on late processes, such as consolidation.

Retrieval: In the subsequent trial, when the same individual is presented again, the spider retrieves the stored memory trace. If the spider recognizes the individual, its behaviour reflects habituation, indicating memory retrieval. Conversely, when a novel individual is introduced, the lack of stored memory trace triggers a different behavioural response, indicating dishabituation. This differential response demonstrates the spider's ability to distinguish between familiar and unfamiliar individuals. This differential response is also key to understanding the nature of habituation over the three sessions, as introducing novel spiders leads to a significant dishabituation response after the three sessions in Experiment 2.

In Line 39, the authors state that they used "a naturalistic experimental procedure". I would like to know how this experiment is 'naturalistic'. The authors' use of an arena does not appear naturalistic, or something the spiders would encounter in the wild.

We appreciate the reviewer's comment regarding our use of the term 'naturalistic'. We acknowledge that the experimental arena itself does not replicate the conditions found in the wild. Our approach aimed to incorporate elements of natural behaviour by allowing two spiders to freely move and interact within the controlled environment. This approach aligns with principles from computational ethology, which seeks to balance the trade-off between repeatability/standardization and observing free, naturalistic behaviour. By using this paradigm, we aimed to capture behaviours that closely resemble those exhibited in their natural habitat. This setup was chosen to balance the need for ecological validity with the requirements for standardized data collection.

Also, and this point has been raised above, by observing the spiders' natural interactions without restraining them or using artificial stimuli like computer animations, we aimed to capture behaviours that closely resemble their natural responses to conspecifics. In contrast, we would not have any clear expectations regarding responses to arbitrarily designed artificial stimuli. This method provides a more ecologically valid assessment of the spiders' recognition abilities.

There are a few details wrong in Line 41. 'Salticidae' is a family name and shouldn't be italicized. Also, the sentence suggests that there is a spider called a 'jumping spider' in the family Salticidae, which is technically called Phidippus regius. To clarify, all spiders in the family Salticidae are known as jumping spiders, and one species of jumping spiders is called Phidippus regius.

We will correct this in the manuscript to accurately reflect the classification and terminology. Thank you for pointing out these inaccuracies.

A manuscript on individual recognition by a salticid should include citations to earlier papers that have already considered individual recognition by salticids. As well as the paper by Tedore and Johnsen (2013), the authors should be aware of the following papers.Clark, R. J., & Jackson, R. R. (1994). Portia labiata, a cannibalistic jumping spider, discriminates between its own and foreign egg sacs. International Journal of Comparative Psychology, 7, 3843.Clark, R. J., & Jackson, R. R. (1994). Self-recognition in a jumping spider: Portia labiata females discriminate between their own draglines and those of conspecifics. Ethology, Ecology & Evolution, 6, 371-375.Clark, R. J., & Jackson, R. R. (1995). Araneophagic jumping spiders discriminate between the draglines of familiar and unfamiliar conspecifics. Ethology, Ecology & Evolution, 7, 185-190.

We appreciate the reviewer's suggestion to include citations to these earlier papers. We will add the recommended references to provide a comprehensive background.

In Line 203, I would not consider "interaction with human caretakers and experimenters" to be a form of behavioral enrichment. This kind of interaction has the potential to be stressful for the spiders, rather than enriching. I suggest deleting that part of the sentence.

We appreciate the reviewer's feedback and agree that interactions with human caretakers and experimenters might not always be enriching and could potentially be stressful for the spiders. We will remove that part of the sentence to better reflect the intended meaning.

**Reviewer #3 (Recommendations For The Authors):**
This manuscript is useful and interesting, and I predict that it will be influential, but more attention should be given to stating the objective and conclusion accurately and clearly. As I understand it, the objective was to investigate a specific hypothesis: that Phidippus regius has a capacity to identify conspecific individuals as particular individuals (i.e., individual identification). Strong evidence supporting this hypothesis being true would be especially remarkable because I am unaware of any published work having shown evidence of a spider expressing this specific perceptual capacity.

Thank you for recognizing the significance and potential influence of our manuscript. We agree that clearly stating the objective and conclusions is essential for conveying the importance of our findings. Our results provide robust evidence supporting the hypothesis that Phidippus regius can recognize and remember individual conspecifics. We will revise the manuscript to more clearly highlight the objective and our conclusions, emphasizing the novel evidence for individual identification in these spiders.

Based on reading this manuscript and based on my understanding of the meaning of 'individual identification', it seems to me that the hypothesis that P. regius has a capacity for individual identification might or might not be true, and the experiments in this manuscript cannot tell us which is the case.

We respectfully disagree with the reviewer's assessment. Our experiments were carefully designed to test whether P. regius has the capacity for individual identification, and our results provide clear evidence supporting this hypothesis. The systematic differences in the spiders' behaviour when encountering familiar versus novel individuals indicate that they can recognize and remember specific conspecifics. We will revise the manuscript to ensure that the evidence and conclusions are stated more clearly to address any potential misunderstandings.

Determining which is the case would have required research that made better use of the literature, and displayed more critical thinking. addressed credible alternative hypotheses and adopted experimental methods that focused more strictly on individual identification.

The distinction between whether P. regius has a capacity for individual identification is not ambiguous in our study. Our findings clearly demonstrate this capacity through systematic behavioural responses to familiar versus novel individuals. As pointed out above, the experimental procedure might be complex, but results are systematic despite this complexity. The experiments were designed to directly address the hypothesis of individual identification, and the data robustly support our conclusions. While considering alternative hypotheses is important, the results we present provide a coherent and compelling case for individual identification in P. regius. We will ensure our manuscript clearly articulates this narrative and the supporting evidence.

At the same time, I also appreciate that asking for all of that at once would be asking for too much. As I see it, this manuscript tells us about research that moves us closer to a clear focus on the details and questions that will matter in the context of considering a hypothesis that is strictly about individual identification. More importantly, I think this research reveals a perceptual capacity that is remarkable even if it is not strictly a capacity for individual identification.

We understand the desire for a more focused exploration of individual identification with paradigms more familiar to the reviewers and we acknowledge that further detailed studies could enhance our understanding of this capacity. However, our findings do indeed suggest that Phidippus regius exhibits a remarkable perceptual capacity for recognizing and remembering individual conspecifics. The systematic behavioural responses observed in our experiments strongly indicate that these spiders possess the ability for individual recognition. While our study may not have explored every potential detail (e.g. which features are most crucial for the memory matching processes), the evidence we present robustly supports the conclusion of individual identification.

We acknowledge that it is indeed valuable to follow established paradigms and build upon the frameworks that have been used successfully in similar species and studies. These paradigms provide a solid foundation for scientific inquiry and allow for comparability across different research efforts. However, it is equally important to acknowledge and explore alternative approaches. Scientific progress is driven not only by replication but also by innovation. By employing new paradigms, researchers can uncover novel insights and push the boundaries of current understanding. The paradigm we used in our study, while different from those traditionally applied to similar research, is not an invention but a well-established method in various domains. It represents an innovative application in the context of our specific research questions, offering a fresh perspective and contributing to the advancement of the field.

As I understand it, 'individual identification' means identifying another individual as being a particular individual instead of a member of a larger set (or 'class') of individuals. An 'individual' is a set containing a single individual. Interesting examples of identifying members of larger sets include discriminating between familiar and unfamiliar individuals. In the context of the specific experiments in this manuscript, familiar-unfamiliar discrimination means discriminating between recently-seen and not-so-recently-seen individuals. My impression is that the experiments in this manuscript have given us a basis for concluding that P. regius has a capacity for familiarunfamiliar (recently seen versus not so recently seen) discrimination. If this is the case, then I think this is the conclusion that should be emphasised. This would be an important conclusion.I appreciate that, depending on how we use the words, familiar-unfamiliar discrimination might be construed as being 'individual identification'. An individual is identified as 'the individual recently seen'. As a casual way of speaking, it can be reasonable to call this 'individual identification'. The difficulty comes from the way calling this 'individual identification' can suggest something more than has been demonstrated. To navigate through this difficulty, we need an expression to use for a capacity that goes beyond familiar-unfamiliar discrimination. In the context of this manuscript about P. regius, we need expressions that will make it easy to consider two things. One of these things is a capacity for familiar-unfamiliar discrimination. The other is the capacity to identify another individual as being a particular individual.

We appreciate the reviewer's insightful comments on the distinction between familiar-unfamiliar discrimination and individual identity recognition. Our study indeed focuses on demonstrating that Phidippus regius can recognize and remember individual conspecifics, providing evidence for individual identity recognition.

Two specific behavioural hallmarks that speak against familiarity recognition:

First, the significant dishabituation response to novel individuals introduced after multiple sessions underscores the specificity of the recognition. This shows that the spiders' habituation is not general but specific to familiar individuals.

Second, the pattern of habituation over the sessions provides further evidence: We observed the strongest systematic modulation in Session 1, a reduced modulation in Session 2, and a further diminished effect in Session 3. If the spiders were only responding based on familiarity, we would expect a more drastic decrease, resulting in a washed-out non-effect by Session 2. However, the continued, though diminishing, differentiation between habituation and dishabituation trials across sessions indicates that the spiders are not merely responding to a general sense of familiarity but are engaging in individual recognition. In other words, the spiders' ability to distinguish between familiar and novel individuals even after repeated exposures suggests that they are not just recognizing a familiar status but are identifying specific individuals.

Things people do might help clarify what this means. People have an extraordinary capacity for identifying other individuals as particular individuals. Often this is based on giving each other names. Imagine we are letting somebody see photographs and asking them to identify who they see. The answer might be, 'somebody familiar' or 'somebody I saw recently' (familiar-unfamiliar discrimination); or the question might be answered by naming a particular individual (individual identification).

We appreciate the reviewer's efforts to clarify the distinction between familiar-unfamiliar discrimination and individual recognition using human examples. However, we believe this comparison might not fully capture the complexity of individual recognition in non-human animals.

Familiarity recognition refers to recognizing someone as having been seen or encountered before without necessarily distinguishing them from others in the same category. On the other hand, identity recognition involves recognizing a specific individual based on unique characteristics (or features). In humans, this often involves naming, but more critically, like in most animals, it involves recognizing visual, auditory, chemical or other sensory cues. In animals, including spiders, individual recognition does not involve and let alone rely on naming but on the ability to distinguish between individuals based on sensory cues and learnt associations. This is a valid and well-documented form of individual recognition across many species.

Individual recognition does not require naming or the assignment of a referential label. Animals can distinguish between specific individuals based on previously perceived and stored features and characteristics. Naming is the exception rather than the rule in the animal kingdom. Only a few species, such as humans and maybe certain cetaceans, use naming for identity recognition. This is an evolutionary rarity and not the standard mechanism for individual recognition, which primarily relies on sensory cues and learnt associations. Furthermore, the mechanism of recognition in both humans and animals involves a complex process of matching incoming sensory and perceptual information with stored memory representations. Naming is merely a tool for communication, allowing us to convey which individual we are referring to. It is not the mechanism by which recognition occurs. The core of individual recognition is this matching process, where sensory cues (visual, auditory, chemical, etc.) are compared to memory traces of previously encountered individuals. Therefore, the suggestion that individual identification necessitates naming misrepresents the actual cognitive processes involved.

We can think of individual identification being based on more fine-grained discrimination (with this, set size = one), with familiar-unfamiliar discrimination being more coarse-grained discrimination (with this, set size can be more than one). Restricting the expression 'individual identification' to instances of having the capacity to identify another individual as being a particular individual (set size = one) is better aligned with normal usage of this expression.

Absolutely, the distinction between fine-grained and coarse-grained discrimination aligns with the concept of different category levels, such as basic and subordinate levels, put forward by Eleanor Rosch (e.g. Rosch, 1973). In the context of individual recognition, fine-grained discrimination (where set size = one) refers to the ability to identify a specific individual based on unique characteristics. This is referred to as subordinate level categorization. Coarse-grained discrimination (where set size can be more than one) refers to recognizing someone as familiar without distinguishing them from others in the same category, more similar to basic level categorization.

Rosch, E.H. (1973). "Natural categories". Cognitive Psychology. 4 (3): 328–50.doi:10.1016/0010-0285(73)90017-0

There is a strong emphasis on an asocial-social distinction in this manuscript. It seems to me that this needs to be focused more clearly on the specific factors that would make a capacity for individual identification beneficial. In the context of this manuscript, the term 'social' may suggest too much. It seems to me that the issue that matters the most is whether individuals live in situations where important encounters occur frequently between the same individuals. Irrespective of whether other notions of the meaning of 'social' also apply, there are salticids that live in aggregated situations where they frequently have important encounters with each other. This is the case with Phidippus regius in the field in Florida, but I realize that there may not be much published information about the natural history of this salticid. Even so, there are salticids to which the word 'social' has been applied in published literature.

We appreciate the reviewer's comments on the asocial-social distinction and we agree that this terminology might need refinement. Our intent was not to categorize Phidippus regius rigidly but to explore the contextual factors influencing the benefits of individual identification. The critical factor in our study is indeed the frequency and importance of encounters between individuals, rather than a broader social structure. We will revise the manuscript to reflect this more nuanced perspective, focusing on the ecological validity of our experimental design and the adaptive significance of individual recognition in environments where repeated encounters can occur.